# Mind the Gap: Catching Hallucinations via Evidence Drop on the Reasoning Manifold

Qunjie Chen[1]   Yufei Chen[1†]   Xiaodong Yue[2]   Linye Li[1]

## Abstract

Large Language Models (LLMs) show strong reasoning abilities, yet their reliability is hindered by hallucinations, where fluent reasoning becomes factually or logically incorrect. Most existing uncertainty-based detectors rely on sequence-level averaging, which ignores the step-wise dynamics of reasoning and often misclassifies hard-but-correct or easy-but-wrong samples. We propose a dynamic perspective that models reasoning as a trajectory on a latent *Evidence Manifold*, where each step is supported by local evidence. Hallucinations are characterized as *Evidence Drops*, i.e., sudden declines in local evidence support that indicate topological deviations from this manifold. Based on this insight, we design a training-free and model-agnostic detector that identifies hallucinations via the worst-case Evidence Drop and enables step-level error localization. Experiments on GSM8K, MATH, and ProcessBench show consistent improvements over sequence-level uncertainty baselines in selective accuracy and riskcoverage trade-offs.

## 1. Introduction

Large Language Models (LLMs) have demonstrated remarkable capabilities in complex reasoning tasks through Chain-of-Thought (Wei et al., 2022). Nevertheless, the reliability of the reasoning process is substantially undermined by the problem of *hallucinations* (Ji et al., 2023; Turpin et al., 2023; Cheng et al., 2025; Shi et al., 2026), in which models produce reasoning that is linguistically coherent yet factually incorrect. To address this challenge, recent studies have explored a wide range of signals for hal-

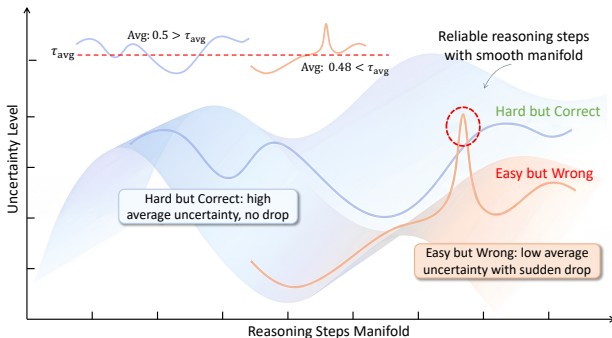

*Figure 1.* Dynamics of LLMs on the proposed latent evidence manifold. Previous methods rely on sequence-level average uncertainty, which incorrectly flags hard but correct samples as hallucinations because their averaged uncertainty exceeds the threshold $\tau_{\text{avg}}$, while overlooking easy but wrong samples whose average uncertainty is low despite exhibiting sharp uncertainty transitions.

lucination detection, including internal hidden states (Yang et al., 2026b; 2025c;d), attention patterns, and output probability distributions (Bazarova et al., 2025; Kim et al., 2025; Zhang et al., 2025; Sriramanan et al., 2024; Orgad et al., 2024). Among them, a predominant line of research focuses on logit-based uncertainty estimation as a principled signal for hallucination detection, leveraging the models intrinsic probabilistic structure to identify unreliable generations (Malinin & Gales, 2020; Farquhar et al., 2024).

However, existing uncertainty-based hallucination detection methods predominantly adopt an *average-over-sequence* paradigm, in which uncertainty is represented as a single sequence-level score obtained by aggregating uncertainty signals across all tokens or reasoning steps, rather than being modeled as a structured, step-wise process. For instance, Length-Normalized Scoring (LN-S) (Malinin & Gales, 2020), Semantic Entropy (Farquhar et al., 2024), and LogTokU (Ma et al., 2025) quantify uncertainty by summarizing token-level or semantic-level statistics over the entire output sequence into a single scalar measure. Such sequence-level aggregation inevitably obscures the fine-grained dynamics of uncertainty evolution during multi-step reasoning and masks how errors emerge and propagate across individual steps (Lightman et al., 2023).

[1]School of Computer Science and Technology, Tongji University [2]Artificial Intelligence Institute, Shanghai University. Correspondence to: Yufei Chen <yufeichen@tongji.edu.cn>.

*Proceedings of the 43rd International Conference on Machine Learning*, Seoul, South Korea. PMLR 306, 2026. Copyright 2026 by the author(s).

In this work, we propose a paradigm shift from static aggregation to dynamic process monitoring. We posit that a valid reasoning process can be viewed as a trajectory evolving on a coherent **Evidence Manifold**, where each intermediate step is supported by a latent evidence state that provides the internal support structure of reasoning: it grounds each step, maintains logical consistency across transitions, and ensures the continuity of the reasoning trajectory. When a hallucination occurs, this continuity is disrupted. The model transitions into a region where the inferred reasoning is no longer sufficiently supported by coherent evidence, leading to a breakdown in the underlying evidence structure. Consequently, a hallucination is not merely a low-probability event or an instance of high uncertainty, but rather a topological deviation from the manifold. We theoretically characterize such deviations as an **Evidence Drop**, defined as a sudden and sharp decline in the models local evidence support between consecutive reasoning steps. Specifically, we quantify local evidence mass using token-level uncertainty signals and detect hallucinations by identifying their abrupt deviations, rather than aggregating uncertainty in an average manner over the entire sequence. To this end, we further integrate our metric with a hypothesis testing framework, which enables us to derive a statistically grounded decision threshold. This framework provides finite-sample guarantees and strictly controls the rate of accepted hallucinations (Type I error) below a user-specified significant level, ensuring reliability in practical deployment. Our main contributions are summarized as follows:

- We model multi-step reasoning as a trajectory on an *Evidence Manifold* and characterize hallucinations as *topological deviations*, identified by sudden local drops in evidence support.

- We propose a training-free, model-agnostic hallucination detector based on the worst-case Evidence Drop, compatible with various uncertainty measures and capable of step-level error localization.

- Experiments on GSM8K, MATH, and ProcessBench demonstrate consistent improvements over sequence-level uncertainty baselines in selective accuracy and riskcoverage trade-offs.

## 2. Related Work

### 2.1. Uncertainty Estimation in LLMs

Quantifying uncertainty in LLMs is a central problem for building reliable and trustworthy systems. Existing approaches can be broadly categorized into logits-based and consistency-based methods. Logits-based methods derive uncertainty directly from the models output distribution, such as token probabilities, entropy, or related scoring functions (Malinin & Gales, 2020; Farquhar et al., 2024; Ma et al., 2025). These approaches are attractive due to their simplicity and their tight connection to the probabilistic formulation of language models. Consistency-based methods, on the other hand, measure uncertainty through variability across multiple sampled generations, such as Self-Consistency (Wang et al., 2022) and its extensions, which interpret disagreement among sampled reasoning paths or answers as a signal of unreliability. However, most existing uncertainty estimation methods ultimately summarize uncertainty into a single scalar score at the sequence level, typically by averaging or aggregating token- or step-wise statistics. While effective for coarse-grained reliability assessment, such static formulations overlook how uncertainty evolves during multi-step reasoning and cannot reveal localized failures within the reasoning process.

### 2.2. Process Monitoring and Step-level Verification

Beyond token-level supervision, a growing body of work has emphasized the importance of monitoring the reasoning process itself. Process Reward Models (PRMs) (Lightman et al., 2023) train separate verifiers to score the correctness of each intermediate step in a reasoning chain, and have demonstrated strong empirical performance in improving reasoning reliability. Related approaches include training step-level critics or verifiers for mathematical and logical reasoning (Cobbe et al., 2021b; Lightman et al., 2023; Wang et al., 2024; Luo et al., 2024), as well as methods that analyze internal activations or hidden states to detect errors (Azaria & Mitchell, 2023). Despite their effectiveness, these methods typically require step-level annotations and additional supervised training, which can be expensive and difficult to scale across domains and tasks.

## 3. The Proposed Method

We formalize the reasoning process as a two-level stochastic system consisting of latent evidence states and observable uncertainty signals. The latent level captures the abstract evidence supporting each underlying reasoning step, while the observable level corresponds to uncertainty-related quantities derived from the models output distribution. This distinction allows us to define hallucinations as topological violations in the latent evidence space that manifest as statistical anomalies in the observation space.

### 3.1. Markov Dynamics on the Evidence Manifold

**Latent Evidence Manifold.** We assume the reasoning process is driven by a sequence of latent evidence states, $\mathbf{E} = (E_1, E_2, \ldots, E_T)$, where each $E_i \in \mathcal{M}$ lies on a low-dimensional manifold $\mathcal{M}$, referred to as the *Evidence Manifold*. The variable $E_i$ represents the abstract internal evi-

dence support associated with the $i$-th reasoning step and $T$ denotes the total number of reasoning steps (or tokens) in the sequence. This assumption is motivated by the manifold hypothesis, which posits that high-dimensional structured data concentrate near low-dimensional manifolds. In our context, $\mathcal{M}$ captures the intrinsic structural constraints that characterize coherent and logically consistent reasoning trajectories. We model the evolution of evidence states as a Markov process on $\mathcal{M}$,

$$P_{\text{true}}(E_1, \ldots, E_T) = P(E_1) \prod_{i=1}^{T-1} P_{\text{true}}(E_{i+1} \mid E_i), \quad (1)$$

which implies that valid reasoning corresponds to a locally smooth transmission of evidence along the manifold. While $E_i$ is latent and unobservable, in practice the models belief at step $t$ is expressed through a predictive distribution with a finite effective support, induced by decoding strategies such as top-$K$ sampling.

**Observation Model: Uncertainty Signals.** The latent evidence states $\{E_i\}$ are not directly observable. Instead, we observe a sequence of uncertainty signals $\mathcal{U} = (U_1, U_2, \ldots, U_T)$, where each $U_i$ is computed from the models output distribution at step $i$. Concretely, $U_i$ may correspond to token-level negative log-likelihood, entropy, or other logit-derived uncertainty measures. We treat all such quantities uniformly as observable realizations of an underlying evidence state. We assume that the uncertainty signal $U_i$ is generated from the latent evidence state $E_i$ through an observation model

$$U_i = g(E_i) + \varepsilon_i, \quad (2)$$

where $g(\cdot)$ is an unknown (possibly nonlinear) observation function and $\varepsilon_i$ represents observation noise. Unlike existing methods that operate directly on $\{U_i\}$, we treat uncertainty as an indirect and noisy observation of an underlying latent evidence process.

**Locality of Evidence.** Inspired by the findings of (Prystawski et al., 2023) that the effectiveness of reasoning comes from the local statistical structure of the training data, we propose that reasoning is fundamentally enabled by the local structure of evidence. In many real-world scenarios, training data does not contain all variables jointly but only presents them in overlapping local neighbors. For example, one subset of observations may capture the relationship *Weather → Road Condition → Commuting Time*, while another subset contains *Road Condition → Traffic Accident → Rescue Time*. Although *Weather* and *Rescue Time* never co-occur in the training data, their relationship can be inferred by chaining accurate local inferences through the shared variable *Road Condition*. This illustrates that global reasoning emerges from the composition of reliable local transitions. We refer to this principle as the ***locality of***

***evidence***: the training data $P_{\text{train}}$ only constrains local transitions between adjacent evidence states, while long-range reasoning is achieved by chaining these locally consistent transitions along a trajectory. Based on this principle, we characterize the training distribution $P_{\text{train}}$ through a *local observation constraint*, which enforces that the model only observe and learn transitions between adjacent evidence states on the latent evidence manifold. This constraint transfers the topological structure of the manifold directly onto the observation space.

**Assumption 1 (Locality of Transitions).** For any two non-adjacent evidence states $E_i$ and $E_j$ with $|i - j| > 1$, the training data contains no direct transitions between them:

$$P_{\text{train}}(U_j \mid U_i) = 0. \quad (3)$$

**Assumption 2 (Local Consistency).** For adjacent evidence states $E_i$ and $E_{i+1}$, the observed transition distribution preserves the true local transition structure on the evidence manifold:

$$P_{\text{train}}(U_{i+1} \mid U_i) \propto P_{\text{true}}(E_{i+1} \mid E_i). \quad (4)$$

### 3.2. Evidence Drop

Modern large language models explicitly regulate the entropy of their predictive distributions to balance exploration and exploitation, prevent premature mode collapse, and ensure training stability. Such entropy control mechanisms have been widely adopted in recent LLMs, including temperature scheduling, entropy penalties, and KL-regularized objectives (e.g., in pretraining or RL stages) (Yang et al., 2025a;b; He et al., 2025; Cui et al., 2025). Accordingly, their learning objective can be abstracted into a generic *entropy-regularized risk minimization* form:

$$R(q) = \underbrace{\mathbb{E}_{P_{\text{train}}}[-\log q(U)]}_{\text{data fitting}} + \lambda \underbrace{\mathbb{E}_{\text{Unif}(\mathcal{V})}[-\log q(U)]}_{\text{entropy regularization}}. \quad (5)$$

where $q(U)$ denotes the models predictive distribution over the uncertainty observation $U$, $\text{Unif}(\mathcal{V})$ is the uniform distribution over the vocabulary (or token space) $\mathcal{V}$, and $\lambda > 0$ controls the strength of entropy regularization. This formulation captures the trade-off between fitting the training data and maintaining sufficient predictive entropy.

Under this setting, we derive the following theorem characterizing the model behavior under topological violations of the evidence manifold.

**Theorem 3.1.** *Let $q^*$ be the optimal estimator minimizing the risk $R(q)$. For any attempt to transition between two non-adjacent evidence states $(E_i, E_k)$ with $|i - k| > 1$*

*(i.e., an off-manifold or topological deviation), the optimal predictive distribution over the uncertainty observation degenerates to the uniform distribution:*

$$q^*(U \mid E_i) = \text{Unif}(\mathcal{V}) = \frac{1}{|\mathcal{V}|}. \tag{6}$$

*Consequently, the corresponding evidence measure collapses to its theoretical minimum:*

$$E_{hallucination} \approx \log\left(\frac{K}{|\mathcal{V}|}\right) \ll 0, \tag{7}$$

*where $K$ denotes the top-$K$ sampling during inference. The full proof is provided in Appendix B.*

Intuitively, Theorem 3.1 states that due to Assumption 1, only transitions between adjacent evidence states are supported by empirical co-occurrence statistics in the training data $P_{\text{train}}$. For such adjacent pairs, the data-fitting term pulls $q$ toward the true conditional distribution, and the entropy regularization term pushes $q$ toward the uniform distribution; the optimal solution for adjacent pairs is therefore a convex combination of the two, yielding a smoothed but still informative predictive distribution. In contrast, for non-adjacent evidence states $(E_i, E_j)$ with $|i - j| > 1$, no joint observations are available under the locality constraint. Consequently, the data-fitting term vanishes, and the risk reduces solely to the entropy regularization term. Minimizing this term forces the optimal predictor to degenerate to the uniform distribution $\text{Unif}(\mathcal{V})$, which represents complete uncertainty and the absence of supporting evidence.

Therefore, an off-manifold transition is intrinsically mapped to a state of maximal entropy and minimal evidential support. This reveals a fundamental asymmetry between valid reasoning and hallucination:

- **On-Manifold (Valid Reasoning).** For adjacent transitions, $P_{\text{train}}(E_{i+1} \mid E_i)$ is nonzero, and the data-fitting term dominates the risk, yielding a concentrated predictive distribution and stable evidence, i.e., $E_{\text{valid}} \approx 0$.

- **Off-Manifold (Hallucination).** For non-adjacent transitions, $P_{\text{train}}$ vanishes due to locality. The risk is governed solely by entropy regularization, forcing the predictor toward the uniform distribution over $\mathcal{V}$ and causing evidence collapse.

As a result, a pronounced ***Evidence Drop*** emerges:

$$\Delta = E_{\text{hallucination}} - E_{\text{valid}} \approx \log K - \log|\mathcal{V}| \ll 0. \tag{8}$$

This result establishes that any off-manifold transition necessarily induces a negative evidence drop, i.e., $\Delta < 0$.

Hence, hallucinations are not identified by low absolute evidence values, but by the occurrence of evidence decreases. Accordingly, we only track positions in the reasoning trajectory where evidence drops appear.

### 3.3. Tracking the Evidence Drop

Guided by Theorem 3.1, our objective is not merely to quantify static uncertainty, but to detect the dynamic phenomenon we term the ***Evidence Drop***. This corresponds to a sharp, non-smooth disruption of the topological support of the latent evidence manifold during the reasoning process. Since the latent evidence states **E** are not directly observable, an Evidence Drop cannot be measured explicitly. Instead, leveraging Assumption 2, we utilize observable uncertainty dynamics as an empirical proxy for the latent evidence manifold. Specifically, we construct a surrogate measure based on the models predictive distribution and track its temporal evolution along the reasoning trajectory.

**1. Entropy as an Empirical Proxy for Latent Evidence.** To make the latent evidence $E_t$ tractable, we adopt Shannon entropy (Shannon, 1948) as our primary uncertainty measure, defining its negative as a proxy for evidence. While alternative metrics exist (e.g., LogTokU (Ma et al., 2025), LN-S (Malinin & Gales, 2020)), they are often heuristic and lack direct alignment with the training objective in Eq. 5. In contrast, Shannon entropy is explicitly coupled with the entropy regularization term in the risk function, admitting a principled interpretation within our theoretical framework.

We define the empirical evidence $\hat{E}_i$ at step $i$ using the next-token predictive distribution. To mitigate the influence of the long-tail vocabulary noise, we compute the entropy over a renormalized top-$K$ distribution. Let $\mathcal{V}_K^{(t)}$ be the set of $K$ tokens with the highest probabilities under $P(\cdot|y_{<t})$. The restricted distribution $\tilde{P}$ is defined as:

$$\tilde{P}(v \mid y_{<i}) = \frac{P(v \mid y_{<i})}{\sum_{v \in \mathcal{V}_K^{(t)}} P(v \mid y_{<i})}. \tag{9}$$

Accordingly, the empirical evidence proxy is:

$$\hat{E}_i := -\mathcal{H}(\tilde{P}(\cdot|y_{<i})) = \sum_{v \in \mathcal{V}_K^{(t)}} \tilde{P}(v|y_{<i}) \log \tilde{P}(v|y_{<i}). \tag{10}$$

By defining $\hat{E}_i$ as the negative entropy, we align it with the physical interpretation of evidence: higher values indicate stronger evidential support (a concentrated distribution), while lower values signify a collapse in certainty.

**2. Identifying the Evidence Drop.** Our core premise is that valid reasoning preserves a smooth evolution of evidence, whereas hallucinations manifest as abrupt structural collapses. To capture this behavior, we analyze the local

gradient of a smoothed evidence trajectory. We first compute a smoothed sequence $\tilde{\mathbf{E}} = \{\tilde{E}_1, \dots, \tilde{E}_T\}$ via an exponential moving average (EMA) over the raw proxy sequence $\hat{\mathbf{E}}$. For each time step $j > 1$, the evidence flux is defined as:

$$\Delta_j = \tilde{E}_j - \tilde{E}_{j-1}. \tag{11}$$

An **Evidence Drop** is characterized by a significant negative gradient, $\Delta_j \ll 0$, indicating a sudden loss of probability mass concentration. To isolate these deleterious events, we consider the set of negative fluctuations $\mathcal{D}^- = \{\Delta_j \mid \Delta_j < 0\}$. Let $\Delta_{(1)} \leq \Delta_{(2)} \leq \dots$ denote the elements of $\mathcal{D}^-$ sorted in ascending order of magnitude. The final risk score $\phi(x, y)$ is formulated by aggregating the $M$ most severe drops:

$$\phi(x, y) = -\frac{1}{M} \sum_{j=1}^{M} \Delta_{(j)}, \tag{12}$$

where $M$ is a hyperparameter (we set $M = 5$ in our experiments). A higher risk score indicates a more pronounced structural violation of the evidence manifold, suggesting a higher likelihood that the reasoning process has deviated into a hallucinated state.

## 4. Experiments

In this section, we conduct extensive experiments to evaluate the effectiveness, reliability, and interpretability of our proposed framework. It is worth noting that our framework operates in a strictly training-free method, requiring no gradient updates or auxiliary reward models. Specifically, we aim to answer the following Research Questions (RQs):

- **RQ1 (Selective Performance):** Does the detection of dynamic *Evidence Drops* yield superior selective prediction performance and a better accuracy-rejection trade-off compared to conventional sequence-averaging uncertainty baselines?

- **RQ2 (Theoretical Alignment):** Is the efficacy of the *Evidence Drop* framework inherently tied to Shannon entropy due to its alignment with the optimization evidence manifold, and does it outperform heuristic uncertainty measures that lack such theoretical coupling?

- **RQ3 (Error Localization):** Can the sharp gradients in the evidence trajectory faithfully pinpoint the specific step where a reasoning deviation (hallucination) occurs within complex Chain-of-Thought traces?

**Datasets.** We evaluate our framework across two distinct granularities of reasoning:

- **Sequence-level Evaluation:** We first utilize GSM8K (Cobbe et al., 2021a) and MATH (Hendrycks

et al., 2021) to assess the model's ability to detect overall reasoning failure. These benchmarks require multi-step logical derivation, where we compute the aggregate uncertainty (or maximum evidence drop) over the entire trajectory to perform selective prediction on the final answer.

- **Step-level Evaluation:** We further employ Process-Bench (Zheng et al., 2025), a specialized diagnostic benchmark encompassing tasks from GSM8K, MATH, OlympiadBench (He et al., 2024) and Omni-MATH (Gao et al., 2024) which provides human-annotated labels for each individual reasoning step, identifying exactly where a logical fallacy occurs. This allows us to evaluate whether the *Evidence Drop* precisely aligns with the specific onset of reasoning errors, moving beyond mere sequence-level correlation to true stepwise localization.

**Implementation Details.** We employ the Qwen3-4B and 8B models (Yang et al., 2025a) as our primary reasoning backbones. All inference experiments are conducted on a cluster of four NVIDIA A100 GPUs (40GB). During the generation process, we utilize greedy decoding with temperature $\tau = 0$, and set the nucleus sampling threshold to $p = 0.95$. The top-$K$ vocabulary constraint is fixed at $K = 20$, which directly corresponds to the hyperparameter $K$ utilized in the evidence estimation in Eq. 10. For the step-level diagnostics on ProcessBench, we adopt a *teacher-forcing* (or forced decoding) approach to extract the predictive distributions along the reasoning traces. Specifically, given a pre-defined reasoning chain, we perform a single forward pass to compute the logit distributions for each token transition. This allows us to accurately map the step-wise Evidence Drop to the specific human-annotated logical segments, ensuring that the uncertainty dynamics are calculated over the exact reasoning path intended for evaluation. Our baselines consist of several uncertainty measures, including Shannon Entropy, LN-S (Malinin & Gales, 2020), and LogTokU (Ma et al., 2025). The detailed computation procedures are provided in Appendix D.

**Selective Prediction and Calibration.** To evaluate the reliability of the uncertainty estimates, we adopt a selective prediction (rejection) framework. A reasoning trajectory is deemed unreliable and subsequently rejected if its risk score $\phi(x, y)$ (or the baseline uncertainty) exceeds a decision threshold $\hat{\tau}$. To determine a statistically principled threshold, we formulate the task as a hypothesis testing problem under the Neyman-Pearson paradigm. We first partition the test data into two disjoint subsets of equal size: a *Calibration Set* ($\mathcal{D}_{cal}$), used to compute $\hat{\tau}$, and an *Evaluation Set* ($\mathcal{D}_{eval}$), used for reporting final performance. On $\mathcal{D}_{cal}$, we compute the distribution of risk scores for samples where the model produces incorrect final answers. Given a

pre-defined significance level $\alpha \in (0, 1)$, we set $\hat{\tau}$ as the $(1 - \alpha)$-quantile of the risk distribution. This procedure ensures that the rejection mechanism provides a controlled level of error detection, allowing for a rigorous comparison between our Evidence Drop metric and baseline uncertainty measures across various operating points. The whole process can be seen in Appendix C.

**Metrics.** We utilize a multi-dimensional evaluation suite to assess the effectiveness of our framework:

- **Selective Accuracy (Acc):** This measures the model's reliability on the subset of queries it chooses to answer. Specifically, for a given threshold $\hat{\tau}$, let $\mathcal{D}_{\text{accepted}} = \{(x, y) \in \mathcal{D} \mid \phi(x, y) \leq \hat{\tau}\}$. Selective Accuracy is defined as:

$$\text{Acc} = \frac{|\mathcal{D}_{\text{accepted}} \cap \mathcal{D}_{\text{correct}}|}{|\mathcal{D}_{\text{accepted}}|}. \quad (13)$$

- **Area Under the Risk-Coverage Curve (AURC):** To evaluate the quality of the uncertainty ranking without dependence on a specific threshold $\hat{\tau}$, we report the AURC (Geifman & El-Yaniv, 2017). The Risk-Coverage curve plots the error rate (Risk) against the fraction of samples accepted (Coverage). A lower AURC indicates that the risk score effectively assigns higher values to incorrect predictions, allowing for a better trade-off between performance and abstention.

- **Step-level Localization Accuracy (SLA):** Evaluated specifically on ProcessBench, this metric assesses the model's ability to pinpoint the exact onset of reasoning failure. We define a successful localization if the first significant *Evidence Drop* (i.e., the first step $t$ where $\Delta_t$ exceeds a step-wise threshold) matches the human-annotated first erroneous step in the CoT trace. This measures the diagnostic precision of our method in complex, multi-step derivations.

### 4.1. Comparison with Sequence-Averaging Baselines (RQ1)

We evaluate the efficacy of the *Evidence Drop* across multiple model scales (4B, 8B) and mathematical reasoning benchmarks. The results, summarized in Table 1, demonstrate a consistent and significant advantage in utilizing dynamic drops over static sequence averages for selective prediction. The experimental results confirm that monitoring the ***Evidence Drop*** is substantially more effective at identifying reasoning errors than traditional sequence-averaging methods.

- **Significant Gains in Selective Accuracy:** Across almost all configurations, the "Drop" variant of a metric

outperforms its "Avg" or "Base" counterpart. For instance, on the MATH dataset with Qwen3-8B ($\alpha = 0.05$), the **Shannon Drop** achieves a selective accuracy of **88.26%**, representing a remarkable **+18.02%** improvement over Shannon Avg (70.24%).

- **Rescuing Weak Baselines:** The most dramatic improvement is observed in **LogTokU**. On GSM8K ($\alpha = 0.05$), LogTokU Avg fails to provide a meaningful decision threshold (yielding near 0% accuracy in early iterations), whereas **LogTokU Drop** recovers this signal to achieve **85.57%** accuracy. This suggests that the *absolute* magnitude of evidence is often less informative than the *sudden loss* of evidence during a reasoning step.

- **Robustness Across Complexity:** While the baseline accuracy on MATH is significantly lower ($\sim59\%$) than on GSM8K ($\sim91\%$), the Evidence Drop maintains high selective accuracy (often $> 80\%$). This indicates that the mechanism effectively filters out hallucinated derivations even when the underlying task difficulty increases.

To understand *why* our method works, we analyze the uncertainty distributions in Figure 2. Separability, The histogram for our method (Figure 2f shows a more distinct separation between valid (green) and hallucinated (red) responses compared to Shannon Entropy (Figure 2c). This confirms our theoretical posit: hallucinations manifest as distinct drops rather than just generically high entropy.

Further, as demonstrated in Table 2, the proposed *Evidence Drop* mechanism consistently achieves lower AURC values than static sequence-averaging baselines, confirming its superior capacity for threshold-free error ranking across diverse model scales and task complexities. In particular, **Shannon Drop** emerges as the most potent error detector, notably reducing the AURC on the MATH benchmark for Qwen3-8B from 288.8 to **190.0**, a significant improvement that underscores the diagnostic value of monitoring dynamic manifold collapses over absolute uncertainty levels. This trend is further mirrored in the **LogTokU Drop** variant, which substantially mitigates the poor ranking performance of its static counterpartfor instance, lowering the MATH-4B AURC from 481.3 to **299.1**thereby validating our premise that sudden disruptions in evidential support are more indicative of reasoning deviations than cumulative probability densities. Collectively, these results signify that aligning uncertainty dynamics with the model's optimization manifold provides a robust and scalable framework for reliable selective prediction in multi-step logical derivations.

*Table 1.* The accuracy performance (%) on GSM8k and MATH datasets. Subscripts denote the performance Drop ($\Delta$) relative to the respective baseline (red for gain, green for loss).

| Model | Dataset | Pretrained | LN-S | LN-S Drop | LogTokU Avg | LogTokU Drop | Shannon Avg | Shannon Drop | $\alpha$ |
|---|---|---|---|---|---|---|---|---|---|
| Qwen3-4B | GSM8K | $87.63 \pm 0.31$ | 95.45 | $83.33_{-12.12}$ | 0 | $85.57_{+85.57}$ | 92.31 | $100_{+7.69}$ | 0.05 |
| | | $87.63 \pm 0.31$ | 90.70 | $89.83_{-0.87}$ | 44.44 | $87.70_{+43.26}$ | 91.53 | $100_{+8.47}$ | 0.10 |
| | | $87.63 \pm 0.31$ | 91.39 | $87.99_{-3.40}$ | 65.48 | $89.17_{+23.69}$ | 90.87 | $90.51_{-0.36}$ | 0.50 |
| | MATH | $59.24 \pm 0.21$ | 73.99 | $82.17_{+8.18}$ | 17.19 | $67.35_{+50.16}$ | 71.43 | $84.94_{+13.51}$ | 0.05 |
| | | $59.24 \pm 0.21$ | 74.94 | $75.76_{+0.82}$ | 24.11 | $74.59_{+50.48}$ | 74.72 | $83.85_{+9.13}$ | 0.10 |
| | | $59.24 \pm 0.21$ | 69.17 | $63.14_{-6.03}$ | 57.04 | $72.44_{+15.40}$ | 68.76 | $72.56_{+3.80}$ | 0.50 |
| Qwen3-8B | Gsm8k | $91.07 \pm 0.26$ | 98.68 | $94.44_{-4.24}$ | 0 | $96.40_{+96.40}$ | 98.57 | $95.51_{-3.06}$ | 0.05 |
| | | $91.07 \pm 0.26$ | 96.95 | $93.48_{-3.47}$ | 66.67 | $92.73_{+26.06}$ | 95.75 | $96.99_{+1.24}$ | 0.10 |
| | | $91.07 \pm 0.26$ | 95.01 | $90.13_{-4.88}$ | 80.15 | $92.27_{+12.12}$ | 97.06 | $93.72_{-3.34}$ | 0.50 |
| | MATH | $59.24 \pm 0.21$ | 69.79 | $86.94_{+17.15}$ | 30.00 | $81.55_{+51.55}$ | 70.24 | $88.26_{+18.02}$ | 0.05 |
| | | $59.24 \pm 0.21$ | 70.85 | $84.69_{+13.84}$ | 42.52 | $81.12_{+38.60}$ | 70.72 | $85.81_{+15.09}$ | 0.10 |
| | | $59.24 \pm 0.21$ | 72.81 | $71.84_{-0.97}$ | 63.41 | $74.91_{+11.50}$ | 72.28 | $77.13_{+4.85}$ | 0.50 |
| Qwen3.5-27B | MATH | 76.00 | 70.37 | $86.59_{+10.59}$ | 68.83 | $93.14_{+17.14}$ | 62.50 | $92.92_{+16.92}$ | 0.05 |

*Table 2.* Comparison of AURC ($\times 1000$) across different uncertainty estimation methods on GSM8K and MATH datasets. Lower is better.

| Model | Dataset | LN-S | | LogTokU | | Shannon | |
|---|---|---|---|---|---|---|---|
| | | Base | **Drop** | Avg. | **Drop** | Avg. | **Drop** |
| Qwen3-4B | GSM8K | 82.7 | 125.7 | 224.2 | **117.2** | 81.1 | **77.1** |
| | MATH | 302.4 | 327.3 | 481.3 | **299.1** | 303.1 | **239.1** |
| Qwen3-8B | GSM8K | 45.4 | 92.8 | 155.8 | **73.8** | 41.9 | 48.3 |
| | MATH | 283.2 | **246.2** | 402.8 | **233.7** | 288.8 | **190.0** |

*Table 3.* Step-level Localization Accuracy (%) on different datasets across multiple model scales.

| Model | Dataset | LN-S | | LogTokU | | Shannon | |
|---|---|---|---|---|---|---|---|
| | | Avg | Drop | Avg | Drop | Avg | Drop |
| Qwen3-4B | GSM8K | 31.24 | 27.94 | 28.76 | 23.19 | 27.94 | **43.42** |
| | MATH | 28.39 | 24.17 | 21.77 | 23.93 | 24.17 | **32.03** |
| | OlympiadBench | 26.22 | 24.93 | 28.57 | 22.62 | 24.95 | **43.06** |
| | Omni-MATH | 23.21 | 23.60 | 26.87 | 22.96 | 23.67 | **38.04** |
| Qwen3-8B | GSM8K | 32.34 | 34.14 | 27.87 | 29.45 | 27.66 | **46.11** |
| | MATH | 26.91 | 25.71 | 21.77 | 20.27 | 24.62 | **32.90** |
| | OlympiadBench | 26.48 | 26.29 | 18.74 | 18.75 | 26.30 | **41.52** |
| | Omni-MATH | 23.35 | 23.33 | 21.10 | 21.15 | 23.40 | **37.04** |

## 4.2. Synergy with Shannon Entropy (RQ2)

The empirical experiments also support our theoretical intuition that the Evidence Drop is most potent when paired with **Shannon Entropy**, which is explicitly aligned with the model's optimization manifold.

- **Optimal Performance with Shannon Entropy:** In the Qwen3-4B GSM8K trials ($\alpha = 0.05$ and $0.10$), **Shannon Drop** achieves a perfect or near-perfect selective accuracy of **100%**. No other uncertainty proxy reaches this ceiling. This validates the premise that because Shannon entropy is coupled with the entropy regularization in the training objective (Eq. 5), its fluctuations are the most reliable indicators of on-manifold versus off-manifold transitions.

- **Comparison with Heuristic Proxies:** While Log-TokU Drop shows massive gains, its absolute performance often lags behind Shannon Drop. Similarly, **LN-S Drop** frequently shows performance degradation (noted by green subscripts) or marginal gains. This is likely because LN-S focuses only on the probability of the *single* most likely token, whereas Shannon Entropy considers the *entire* distribution, capturing the subtle probability mass shifts that precede a

structural collapse in reasoning.

- **Consistency across Confidence Levels:** Shannon Drop remains the most stable metric as the rejection threshold $\alpha$ varies. Even at $\alpha = 0.50$, where a larger portion of the distribution is accepted, Shannon Drop consistently provides positive gains on the MATH dataset, whereas LN-S Drop often falls below its own static baseline.

## 4.3. Stepwise Error Localization Ability (RQ3)

We further evaluate stepwise error localization using Step-level Localization Accuracy (SLA), which assesses whether an uncertainty signal can precisely identify erroneous reasoning steps, rather than merely detecting unreliable samples at the sequence level. As shown in Table 3, the *Drop*-based criterion consistently outperforms the sequence-level *Avg* criterion across all datasets and uncertainty measures on Qwen3-8B. Notably, Shannon entropy combined with Evidence Drop achieves the best performance on every benchmark. For instance, SLA improves from 27.66% to 46.11% on GSM8K, from 24.62% to 32.90% on MATH, from 26.30% to 41.52% on Olympiad-

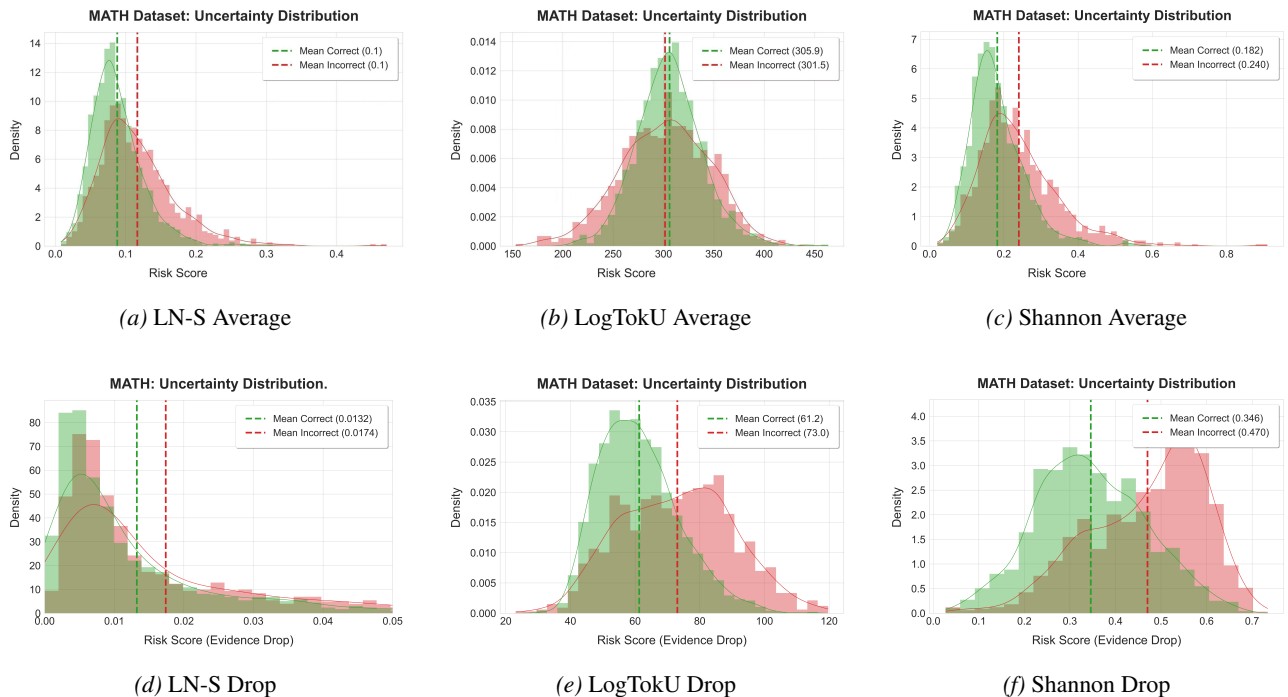

*(a)* LN-S Average        *(b)* LogTokU Average        *(c)* Shannon Average

*(d)* LN-S Drop        *(e)* LogTokU Drop        *(f)* Shannon Drop

*Figure 2.* **Comparison of Uncertainty Distributions.** The top row displays sequence-level aggregation methods (Average), while the bottom row displays our proposed step-level monitoring methods (Drop). The Drop metrics exhibit a clearer separation between correct and hallucinated responses.

Bench, and from 23.40% to 37.04% on Omni-MATH. These results indicate that abrupt uncertainty transitions provide a much stronger signal for step-level error localization than the absolute magnitude of uncertainty.

In contrast, the *Avg*-based criterion shows limited discriminative power. While high average uncertainty often reflects task difficulty, it fails to capture the precise moment when the reasoning trajectory deviates from the evidence manifold, thereby diluting critical local anomalies. The superior performance of the *Drop*-based methods confirms our hypothesis that hallucinations are characterized by nonsmooth disruptions in the uncertainty trajectory, and that effective error localization requires dynamic process monitoring rather than static uncertainty aggregation.

Further, Figure 3 provides a fine-grained comparison between Entropy-Avg and Entropy-Drop. Entropy-Avg exhibits substantial overlap between correct and erroneous steps, showing that averaged entropy cannot reliably distinguish local reasoning failures. In contrast, Entropy-Drop achieves clear separation: erroneous steps correspond to pronounced entropy drops, while correct steps maintain smooth and stable uncertainty trajectories. This demonstrates that it is the *dynamic change* in uncertainty, rather than its static value, that serves as an effective indicator for precise step-level error localization, further validating the Evidence Manifold perspective.

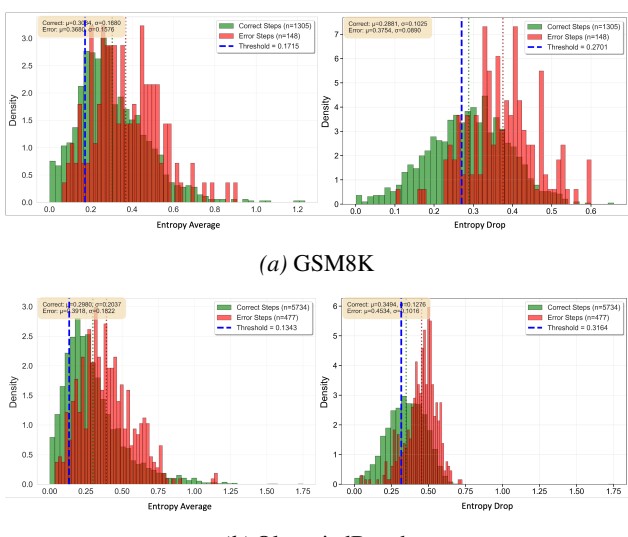

*(a)* GSM8K

*(b)* OlympiadBench

*Figure 3.* Comparison of reasoning-correct and reasoning-incorrect steps on the GSM8K and OlympiadBench datasets using Qwen3-8B, evaluated by entropy-based uncertainty with sequence-level averaging (Entropy-Avg) and stepwise Evidence Drop (Entropy-Drop). The blue dashed line indicates the decision threshold, obtained on the calibration set with significance level $\alpha = 0.1$.

# 5. Conclusion

In this paper, we introduced a dynamic perspective on hallucination detection by modeling reasoning as a trajectory on an ***Evidence Manifold***. We demonstrated that hallucinations are not merely instances of high uncertainty but are characterized by ***Evidence Drops***. Extensive experiments on benchmarks such as GSM8K and MATH show that our training-free, model-agnostic detector significantly outperforms sequence-level averaging baselines in both selective accuracy and step-level error localization. By integrating a hypothesis testing framework, we provide formal statistical guarantees to ensuring reliable deployment in complex reasoning tasks. Finally, the ***Evidence Drop*** mechanism offers an interpretable pathway toward monitoring and securing the integrity of multi-step reasoning in LLMs.

# 6. Limitation

Despite the effectiveness of our Evidence Drop mechanism in identifying unreliable reasoning steps, a primary limitation of this work is its current focus on passive detection rather than active intervention.Given our method's high precision in step-level error localization, future research could extend this signal into an online mitigation system.

## Acknowledgements

This work was supported by the National Natural Science Foundation of China ( Nos. 62472315, 62476165).

## Impact Statement

This paper presents work whose goal is to advance the field of Machine Learning, specifically in enhancing the reliability and safety of Large Language Models (LLMs) during complex reasoning tasks. By introducing the Evidence Drop mechanism and providing rigorous statistical guarantees under a hypothesis testing framework, our method facilitates the safe deployment of LLMs in high-stakes domainssuch as medical diagnosis, financial analysis, and legal reasoningwhere logical hallucinations can lead to severe real-world consequences. Furthermore, as a training-free and model-agnostic approach, our framework significantly reduces the computational overhead and carbon footprint typically associated with training specialized process reward models or verifiers. We believe that improving the transparency and interpretability of AI systems through step-level error localization fundamentally aligns with the broader societal goals of developing trustworthy, responsible, and aligned Artificial Intelligence. There are no obvious adverse ethical or societal consequences expected from this work.

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

# A. Symbols

*Table 4.* Summary of Notations and Definitions used in this paper.

| Symbol | Description & Physical Meaning |
|---|---|
| *Manifold & Theoretical Framework* | |
| $\mathcal{M}$ | The low-dimensional **Evidence Manifold** representing valid reasoning trajectories. |
| $E_i$ | **Latent Evidence State** at step $i$. Represents the intrinsic logical support. |
| $U_i$ | **Observed Uncertainty Vector** (e.g., logits) at step $i$. The tangible observation of $E_i$. |
| $P_{\text{true}}$ | The **True Distribution** on the manifold (Ideal Logic). |
| $P_{\text{true}}(v)$ | The probability of generating a specific token $v$ on the ideal manifold. *Note:* The specific geometry is determined by the chosen uncertainty metric. |
| $P_{\text{train}}$ | The **Training Distribution** learned by the model. |
| $\Delta_j$ | **Evidence Drop**. The quantitative decline of $\tilde{E}_j$ relative to $\tilde{E}_{j-1}$, indicating a hallucination. |
| *Model Inputs & Calculation* | |
| $y_{<t}$ | **Token History** (Context). The sequence of tokens generated prior to step $t$. |
| $p(v \mid y_{<t})$ | **Next-Token Probability**. The model's predicted probability for token $v$ given history $y_{<t}$. |
| $\hat{E}_i$ | **Calculated Evidence Value** at step $i$ (e.g., Shannon Entropy). It is a scalar realization of $E_i$. |
| $\tilde{E}_i$ | **Smoothed Evidence Value**. The EMA-smoothed version of $\hat{E}_i$ to mitigate noise. |
| $\mathcal{V}$ | The model's complete vocabulary size. |
| *Evaluation & Datasets* | |
| $\phi(x, y)$ | **Sequence-Level Uncertainty Score**. A function (like LN-S) evaluating the quality of the entire response $y$. |
| $\mathcal{D}$ | The dataset containing correct/valid reasoning samples (Positive samples). |
| $\mathcal{D}^-$ | The **Hallucination Dataset** containing incorrect/fabricated samples (Negative samples). |
| $K$ | **Hyperparameter**. Top-K candidates used for evidence approximation (e.g., $K = 20$). |

# B. Theoretical analysis

In this section,we will provide a detailed proof of our theoretical framework.

## B.1. Problem Setup

**Evidence Manifold and Factorization**

We assume that the reasoning process of an LLM is driven by a series of latent **Evidence States** $\mathbf{E} = (E_1, E_2, \ldots, E_T)$,, which exist on a low-dimensional data manifold $\mathcal{M}$. Here, $E_i$ can not only represent an abstract logical step, but also be specifically quantitied by different uncertainty measures.

We assume that the true evidence distribution defined on the manifold satisfies the Markov Property. That is, the joint distribution of the entire evidence chain can be factorized into the product of local conditional probabilities:

$$P_{\text{true}}(E_1, \ldots, E_N) = P(E_1) \prod_{t=1}^{N-1} P_{\text{true}}(E_{t+1} \mid E_t) \tag{14}$$

This formula implies that a valid reasoning process is a local transmission of evidence on the manifold, where the evidence state $E_{t+1}$ at each step only depends on the preceding step $E_t$.

**Observation Model: Indices and Uncertainty Measurement.** To capture these latent evidence states, we define the observation data as serialized index-observation pairs. For each step in the reasoning, the model outputs a tuple $(i, U_i)$, where $i$ is the step index and $U_i$ is the corresponding vector (e.g., logits or probability distribution) from which uncertainty is measured. The latent evidence state $E_i$ can be derived from $U_i$, i.e., $E_i = f(U_i)$.

**Local Observation Assumptions.** We define the distributional characteristics of the training data $P_{\text{train}}$ through the following two topological assumptions:

**Assumption 1 (Locality of Transitions).** For any two non-adjacent evidence states $E_i$ and $E_j$ with $|i-j| > 1$, the training data contains no direct transitions between them:

$$P_{\text{train}}(U_j \mid U_i) = 0. \tag{15}$$

This implies that a direct skip across the manifold constitutes an Out-of-Distribution (OOD) event.

**Assumption 2 (Local Consistency).** For adjacent evidence states $E_i$ and $E_{i+1}$, the observed transition distribution preserves the true local transition structure on the evidence manifold:

$$P_{\text{train}}(U_{i+1} \mid U_i) \propto P_{\text{true}}(E_{i+1} \mid E_i). \tag{16}$$

This implies that the model learns to faithfully approximate the local evidence transmission logic.

### B.2. Problem Transformation

**Proposition 1**. Let $\alpha_{\text{data}} \geq 0, \alpha_{\text{reg}} \geq 0$. Define the risk function $R(q)$ as the weighted sum of a data fitting term and a regularization term:

$$R(q) = \alpha_{\text{data}} \mathbb{E}_{P_{\text{train}}(U)}[-\log q(U)] + \alpha_{\text{reg}} \mathbb{E}_{\mathcal{U}(U)}[-\log q(U)] \tag{17}$$

where $U$ is the observed values vector, and $\mathcal{U}$ is the uniform distribution. Then, the optimal distribution $q^* = \arg\min_q R(q)$ that minimizes the risk satisfies:

$$q^*(U) = \frac{\alpha_{\text{data}}}{\alpha_{\text{data}} + \alpha_{\text{reg}}} P_{\text{train}}(U) + \frac{\alpha_{\text{reg}}}{\alpha_{\text{data}} + \alpha_{\text{reg}}} \mathcal{U}(U) \tag{18}$$

*Remark:* The risk function formulated in Eq. (5) of the main text corresponds to a special case of this general objective where we set the weights to $\alpha_{\text{data}} = 1$ and $\alpha_{\text{reg}} = \lambda$.

**Proof**. We assume the probability distribution is discrete (defined on a finite vocabulary $\mathcal{V}$). To solve for the extremum under the constraint $\sum_U q(U) = 1$, we construct the Lagrangian function:

$$\mathcal{L}(q, \lambda_0) = -\alpha_{\text{data}} \sum_U P_{\text{train}}(U) \log q(U) - \alpha_{\text{reg}} \sum_U \mathcal{U}(U) \log q(U) + \lambda_0 \left( \sum_U q(U) - 1 \right) \tag{19}$$

The first-order conditions with respect to $q(U)$ and $\lambda_0$ are:

$$\frac{\partial \mathcal{L}}{\partial q(U)} = -\alpha_{\text{data}} \frac{P_{\text{train}}(U)}{q(U)} - \alpha_{\text{reg}} \frac{\mathcal{U}(U)}{q(U)} + \lambda_0 = 0 \tag{20}$$

$$\frac{\partial \mathcal{L}}{\partial \lambda_0} = \sum_U q(U) - 1 = 0 \tag{21}$$

From Eq. (20), we obtain the expression for $q(U)$:

$$q(U) = \frac{\alpha_{\text{data}} P_{\text{train}}(U) + \alpha_{\text{reg}} \mathcal{U}(U)}{\lambda_0} \tag{22}$$

To solve for the normalization constant $\lambda_0$, we substitute the above expression into the normalization constraint (21):

$$\sum_U \frac{\alpha_{\text{data}} P_{\text{train}}(U) + \alpha_{\text{reg}} \mathcal{U}(U)}{\lambda_0} = 1 \tag{23}$$

Rearranging gives:

$$\frac{1}{\lambda_0} \left( \alpha_{\text{data}} \underbrace{\sum_U P_{\text{train}}(U)}_{1} + \alpha_{\text{reg}} \underbrace{\sum_U \mathcal{U}(U)}_{1} \right) = 1 \tag{24}$$

Since $P_{\text{train}}$ and $\mathcal{U}$ are both valid probability distributions, their sums are 1, thus yielding:

$$\lambda_0 = \alpha_{\text{data}} + \alpha_{\text{reg}} \tag{25}$$

Substituting $\lambda_0$ back into the expression for $q(U)$ yields the stated conclusion. $\qquad\square$

### B.3. Theorem Statement

**Theorem 1: Evidence Collapse from Manifold Deviation**

**Preliminaries** Let $\mathcal{U}$ be the uniform distribution defined on the vocabulary $\mathcal{V}$. Let $P_{\text{train}}$ be the joint distribution over evidence state indices $i$ and observed values $U$ defined by the "Local Observation Constraint" in Section B.1. Let $H(p, q)$ denote the cross-entropy between distributions $p$ and $q$. We consider the following risk function with entropy regularization:

$$R(q) = H(P_{\text{train}}, q) + \lambda H(\mathcal{U}, q) \tag{26}$$

where $\lambda > 0$ is the regularization coefficient.

**Theorem Conclusion** The optimal estimator $q^* = \arg\min_q R(q)$ that minimizes the risk satisfies the following properties and exhibits a significant **"Evidence Drop"**:

- **For all pairs of adjacent evidence states** $(E_i, E_{i+1})$ **on the manifold:**
  The optimal predictive distribution $q^*(U \mid E_i)$ is a weighted mixture of the true manifold distribution and the uniform distribution:

  $$q^*(U \mid E_i) = \alpha P_{\text{true}}(U \mid E_i) + (1 - \alpha)\mathcal{U}(U) \tag{27}$$

  where $\alpha \in (0, 1)$ is a weight close to 1. In this case, the observed evidence quantity $E$ remains in a high-confidence interval:

  $$E(q^*) \approx E(P_{\text{true}}) \approx 0 \tag{28}$$

- **For all pairs of non-adjacent evidence states** $(E_i, E_k)$ **(where $|i - k| > 1$):**
  The optimal predictive distribution $q^*(U \mid E_i)$ strictly degenerates to the uniform distribution:

  $$q^*(U \mid E_i) = \mathcal{U}(U) = \frac{1}{|\mathcal{V}|} \tag{29}$$

  In this case, the observed evidence quantity $E$ undergoes a drastic collapse, reaching the theoretical lower bound:

  $$E(q^*) = \log\left(\frac{K}{|\mathcal{V}|}\right) \ll 0 \tag{30}$$

**Corollary** Therefore, when the model attempts to perform a non-adjacent jump, the variation in evidence value (i.e., the Evidence Drop) $\Delta$ is mathematically bounded as:

$$\Delta = E_{\text{hallucination}} - E_{\text{valid}} \approx \log\left(\frac{K}{|\mathcal{V}|}\right) \ll 0 \tag{31}$$

**Proof.**

### 1. Orthogonal Decomposition of Risk

Using the Law of Iterated Expectations, we decompose the total risk $R(q)$ into a sum over evidence state transition pairs $(i, k)$. According to the Local Observation Constraint defined in Section B.1, we partition the summation domain into **adjacent** ($\mathcal{Z}_{\text{adj}}$) and **non-adjacent** ($\mathcal{Z}_{\text{non}}$) sets:

$$R(q) = \mathbb{E}_{P_{\text{train}}}[-\log q] + \lambda \mathbb{E}_{\mathcal{U}}[-\log q] \tag{32}$$

$$= \sum_{(i,k) \in \mathcal{Z}_{\text{adj}}} \Bigg( \underbrace{P_{\text{train}}(i, k) H(P_{\text{true}}, q) + \lambda H(\mathcal{U}, q)}_{\text{Term I: Visible Transitions}} \Bigg)$$

$$+ \sum_{(i,k) \in \mathcal{Z}_{\text{non}}} \Bigg( \underbrace{0 \cdot H(\cdot, q) + \lambda H(\mathcal{U}, q)}_{\text{Term II: Invisible Transitions}} \Bigg) \tag{33}$$

Note that in Term II, since $P_{\text{train}}(i, k) = 0$, the data fitting term vanishes, and the risk is constituted solely by the regularization term.

## 2. Extremal Analysis

We solve for the optimal estimator $q^* = \arg \min_q R(q)$ for the two cases described above.

**Case A: On-Manifold Adjacent Transitions** $((i, k) \in \mathcal{Z}_{\text{adj}})$

In this case, the local risk function is $r(q) = \alpha H(P_{\text{true}}, q) + \lambda H(\mathcal{U}, q)$. Constructing the Lagrangian $\mathcal{L} = r(q) + \gamma(\sum q - 1)$ and solving for $\nabla_q \mathcal{L} = 0$:

$$-\frac{\alpha P_{\text{true}}(U)}{q(U)} - \frac{\lambda \mathcal{U}(U)}{q(U)} + \gamma = 0 \implies q^*(U) \propto \alpha P_{\text{true}}(U) + \lambda \mathcal{U}(U) \tag{34}$$

Since $\alpha \gg \lambda$ on the training manifold, the optimal solution is dominated by the true distribution:

$$q^*(U \mid E_i) \approx P_{\text{true}}(U \mid E_i) \tag{35}$$

Analyzing the first term (Term I) of the risk decomposition in Eq. (32) (i.e., the adjacent transitions), and substituting the derived approximation $q^* \approx P_{\text{true}}$ into the evidence definition, we obtain:

$$E_{\text{valid}} = \log \left( \sum_{v \in \text{Top-K}(q^*)} q^*(v) \right) \approx \log \left( \sum_{v \in \text{Top-K}(P_{\text{true}})} P_{\text{true}}(v) \right) \tag{36}$$

Since valid reasoning steps on the manifold typically exhibit low uncertainty (i.e., the probability mass is highly concentrated on a few correct tokens), the cumulative probability of the top-$K$ candidates approaches 1:

$$\sum_{v \in \text{Top-K}(P_{\text{true}})} P_{\text{true}}(v) \approx 1 \implies E_{\text{valid}} \approx \log(1) = 0 \tag{37}$$

**Case B: Off-Manifold Non-Adjacent Jumps** $((i, k) \in \mathcal{Z}_{\text{non}})$

Here, the data term weight $\alpha = 0$, and the local risk function degenerates to $r(q) = \lambda H(\mathcal{U}, q)$. The optimal solution must minimize the cross-entropy with the uniform distribution:

$$\nabla_q[\lambda H(\mathcal{U}, q)] = 0 \implies q^*(U \mid E_i) = \mathcal{U}(U) = \frac{1}{|\mathcal{V}|} \tag{38}$$

This represents a perfectly flat distribution. The Top-K evidence $E_{\text{hallucination}}$ is thus:

$$E_{\text{hallucination}} = \log \left( \sum_{j=1}^{K} \frac{1}{|\mathcal{V}|} \right) = \log \left( \frac{K}{|\mathcal{V}|} \right) \tag{39}$$

## 3. Evidence Drop Bound

Defining the evidence variation $\Delta = E_{\text{hallucination}} - E_{\text{valid}}$, and based on the results above:

$$\Delta = \log \left( \frac{K}{|\mathcal{V}|} \right) - E_{\text{valid}} \tag{40}$$

$$\leq \log \left( \frac{K}{|\mathcal{V}|} \right) - (1 - \epsilon) \approx \log K - \log |\mathcal{V}| \ll 0 \tag{41}$$

where $|\mathcal{V}|$ is the vocabulary size (typically $> 10^4$) and $K$ is a constant. Therefore, $\Delta$ is a significant negative value.

$\square$

# C. Hypothesis Testing

In this part,we will elaborate in detail on how to obtain statistically guaranteed thresholds through hypothesis testing.

## C.1. Statistical Guarantees in Machine Learning

Reliable deployment of machine learning systems often requires rigorous statistical guarantees beyond empirical performance. Conformal prediction (Angelopoulos & Bates, 2023) provides distribution-free guarantees on coverage by constructing prediction sets, while risk control frameworks (Bates et al., 2021) bound expected loss or error under specified constraints. These approaches have been widely adopted in classification and multi-view classification (Angelopoulos et al., 2020; Bates et al., 2021; Liu et al., 2025a;b; 2024), regression (Lee et al., 2009; Barber et al., 2021; Yang et al., 2026a), and selective prediction settings (Geifman & El-Yaniv, 2017; Dhillon et al., 2024). In parallel, hypothesis testing offers a principled mechanism for controlling error rates in binary decision problems, such as the False Positive Rate or Type I error, with finite-sample guarantees (Tong et al., 2016). In this work, we formulate hallucination detection as a hypothesis testing problem, where accepting a hallucinated reasoning trace corresponds to a Type I error. By integrating our Evidence Drop statistic with a NeymanPearson style decision rule, we obtain explicit finite-sample control over the false acceptance rate.

## C.2. Hypothesis Testing

We deploy the derived risk metric $\phi(x, y)$ ,which quantifies the severity of evidence drops,to construct a binary decision rule. Unlike empirical thresholds, we formulate this as a hypothesis testing problem under the Neyman-Pearson paradigm to provide a finite-sample statistical guarantee against hallucinations:

$$H_0 : \text{The reasoning chain deviates from } \mathcal{M} \text{ (Hallucination).}$$
$$H_1 : \text{The reasoning chain is valid on } \mathcal{M} \text{ (Reliable).}$$

(42)

We aim to control the **Type I Error** (accepting a hallucination) at a user-specified significance level $\alpha$ (e.g., $\alpha = 0.05$). **Calibration and Statistical Bound.** To determine the critical decision threshold $\tau$, we utilize a calibration set of known hallucinations, $\mathcal{D}_{cal} = \{(x_i, y_i) | y_i \text{ is incorrect}\}$. We compute the risk scores $\mathcal{S}_{cal} = \{\phi(x_i, y_i)\}$ for these failure cases. Since higher $\phi$ values indicate severe evidence drops (high risk), a hallucination is falsely accepted if its risk score falls *below* the threshold $\tau$. We seek the smallest $\tau$ such that the probability of this error is bounded:

$$\mathbb{P}(\phi(x, y) \leq \tau \mid H_0 \text{ is true}) \leq \alpha \tag{43}$$

To ensure this bound holds for finite samples without assuming a parametric distribution (e.g., Gaussian), we employ the concentration inequality based on the Binomial tail. Motivated by (Tong et al., 2018),we select $\hat{\tau}$ as the $(1 - \alpha)$-th quantile of the empirical distribution of $\mathcal{S}_{cal}$, adjusted for finite sample size $N_{cal}$ to satisfy:

$$\sum_{j=k}^{N_{cal}} \binom{N_{cal}}{j} (1 - \alpha)^j \alpha^{N_{cal} - j} \leq \delta \tag{44}$$

where $\delta$ is the confidence level of the guarantee itself. **The Decision Rule.** With the calibrated threshold $\hat{\tau}$, the final deployment policy is:

$$\mathrm{D}(y) = \begin{cases} \textbf{Accept} & \text{if } \phi(x, y) \leq \hat{\tau} \quad \text{(Safe Manifold Trajectory)} \\ \textbf{Reject} & \text{if } \phi(x, y) > \hat{\tau} \quad \text{(Detected Deviation)} \end{cases} \tag{45}$$

D(y) is equivalent to Decision(y). This mechanism ensures that, with probability $1 - \delta$, the rate of accepting hallucinations is strictly upper-bounded by $\alpha$, providing a rigorous safety certificate for the LLM's reasoning process.

# D. Baselines

To ensure a rigorous comparison, we implement the following uncertainty baselines:

- **Shannon Entropy:** Following common practice for high-cardinality vocabularies, we compute a computationally efficient approximation using the top-$K$ logits ($K = 20$). For each step $t$, we re-normalize the probabilities of the top-$K$ logits to form a local distribution $\tilde{P}(v|y_{<t})$. The step-wise entropy is

$$H_i = - \sum_{v \in \mathcal{V}_K} \tilde{P}(v|y_{<i}) \log \tilde{P}(v|y_{<i}). \tag{46}$$

  The sequence-level uncertainty score is the mean entropy over the trajectory of length $T$: $\phi_{\text{Shannon}} = \frac{1}{T} \sum_{i=1}^{T} H_i$.

- **LogTokU ([Ma et al., 2025](#)):** This baseline measures the *Evidence Mass* $M_t$, defined as the sum of log-probabilities for the top-$K$ candidates:

$$M_i = \sum_{v \in \mathcal{V}_K} \log P(v|y_{<i}). \tag{47}$$

  High mass indicates a concentration of confidence among the top candidates. The final risk score is the negative average mass: $\phi_{\text{LogTokU}} = -\frac{1}{T} \sum_{i=1}^{T} M_i$.

- **LN-S ([Malinin & Gales, 2020](#)):** This baseline utilizes the Length-Normalized Scoring of the generated sequence. Under greedy decoding (sampling temperature $\tau = 0$), the model selects the token $y_t$ with the maximum likelihood at each step. We compute the average negative log-probability:

$$\phi_{\text{LN-S}} = -\frac{1}{T} \sum_{i=1}^{T} \log P(v|y_{<i}), \tag{48}$$

  where $P(v|y_{<i})$ represents the probability of the token actually generated at step $t$.

# E. Supplementary experiments

### E.1. Hyperparameter Robustness

The Table 5 below present our ablation of our drop methods on the MATH dataset with Qwen3-8B. It is important to note that the numbers reported in the tables represent the selective accuracy (Acc). These results are achieved by rejecting unreliable reasoning trajectories using a statistically guaranteed decision threshold, which is rigorously derived via our hypothesis testing framework at a significance level of $\alpha = 0.05$. In each experiment, we isolate the effect of a single hyperparameter by fixing the other two to our default configuration (i.e., Max Drops $M = 5$, EMA span = 5, and Top-$K = 20$).

For a direct comparison, we include our strongest baseline (Shannon Avg, 70.24%) in our paper to demonstrate the robustness of our Drop method. This serves as the reference baseline for all ablation results below.

*Table 5.* Ablation study of Evidence Drop on the MATH dataset (Qwen3-8B). The reference baseline (Shannon Avg) achieves a selective accuracy of 70.24%. Default settings are $M = 5$, EMA span = 5, and $K = 20$.

**(a) Effect of Max Drops $M$ (EMA span = 5)**

| $M$ | LN-S Drop | LogTokU Drop | Shannon Drop |
|---|---|---|---|
| 1 | 84.58 | 73.40 | 87.26 |
| 3 | 85.77 | 81.56 | 87.92 |
| 5 (Ours) | **86.94** | 81.55 | 88.26 |
| 10 | 85.93 | **83.33** | **90.75** |
| All | 55.43 | 65.66 | 74.81 |

**(b) Effect of EMA Span ($M = 5$)**

| EMA Span | LN-S Drop | LogTokU Drop | Shannon Drop |
|---|---|---|---|
| 1 | 84.58 | 73.40 | 87.26 |
| 3 | 85.77 | 81.56 | 87.92 |
| 5 (Ours) | **86.94** | 81.55 | 88.26 |
| 10 | 85.93 | **83.33** | **90.75** |
| All | 55.43 | 65.66 | 74.81 |

**(c) Effect of Top-$K$ (EMA span = 5, $M = 5$)**

| Top-$K$ | LN-S Drop |
|---|---|
| $K = 2$ | 86.05 |
| $K = 5$ | 85.42 |
| $K = 10$ | **87.64** |
| $K = 20$ (Ours) | 86.94 |

Since $K$ has minimal impact, we use LN-S Drop as a representative example. Following Ma et al. (Ma et al., 2025), we default to $K = 20$. Table 5(c) demonstrates high robustness across $K \in [2, 20]$, proving our method captures intrinsic dynamics without sensitive tuning.

The results demonstrate exceptional hyperparameter robustness. Across $M$, EMA span, and Top-$K$, performance remains highly stable within broad, non-extreme windows. Degradation occurs only in extreme settings that inherently violate our theoretical premises (e.g., $M \to$ All dilutes local signals via global averaging).

### E.2. Comparison with other methods

To compare against generation-based uncertainty methods like Semantic Entropy, we evaluated Self-Consistency (SC), which shares the exact same foundational mechanism: sampling multiple reasoning paths to measure output consistency/entropy. We tested SC with $N = 5, 10,$ and $15$ sampling paths on the MATH dataset. For clarity, the SC results represent the final accuracy after majority voting, which serves as a direct comparison against the baseline accuracy of the pretrained model with zero voting overhead.

*Table 6.* Performance comparison of Self-Consistency (SC) on the MATH dataset. The results indicate the accuracy (%) of answers after majority voting.

| Voting Paths ($N$) | Qwen3-8B | | Qwen3-4B | |
|---|---|---|---|---|
| | Pretrained | + SC | Pretrained | + SC |
| $N = 5$ | 66.12 | 69.76 (+3.64) | 57.92 | 63.72 (+5.80) |
| $N = 10$ | 66.40 | 70.72 (+4.32) | 58.56 | 65.36 (+6.80) |
| $N = 15$ | 65.96 | 70.56 (+4.60) | 57.88 | 65.04 (+7.16) |

**Analysis:** As shown in Table 6, while multi-path methods like SC incur a $15\times$ computational overhead for marginal gains (e.g., reaching 70.56% on Qwen3-8B), our single-path Shannon Drop method avoids these massive generation costs entirely. With just a single forward pass, it delivers a dramatic 10–20% absolute gain, reaching an impressive 88.26%.

### E.3. Generalization to Daily Life Topics

To demonstrate how our method detects daily life topic hallucinations, we conducted new experiments on the ProofWriter dataset.

ProofWriter is a standard benchmark for deductive logical reasoning constructed entirely from natural language statements about everyday entities and scenarios. It perfectly simulates daily-life factual logic.

The results are presented in Table 7. We evaluate the Area Under the Risk-Coverage curve (AURC, where lower is better) on the ProofWriter dataset using Qwen3-8B, covering both Open World Assumption (OWA) and Closed World Assumption (CWA) at depth-3. The experimental settings remain consistent with our main paper.

*Table 7.* AURC (lower is better) on the ProofWriter dataset (Qwen3-8B) for Open World Assumption (OWA) and Closed World Assumption (CWA) at depth-3. The best results for each setting are highlighted in bold.

| Setting | Aggregation | LN-S | LogTokU | Shannon |
|---|---|---|---|---|
| OWA (Depth-3) | Average | 711.82 | 685.96 | 717.17 |
| | Drop (Ours) | 801.13 | **648.28** | **589.02** |
| CWA (Depth-3) | Average | 496.73 | 579.03 | 484.33 |
| | Drop (Ours) | 729.19 | **550.80** | **418.96** |

**Analysis:** As shown in Table 7, our Evidence Drop method utilizing Shannon Entropy as the uncertainty measure achieves the best performance. It significantly outperforms the global averaging baselines by yielding the lowest AURC scores under both OWA and CWA settings, demonstrating its strong generalization capabilities in daily-life factual logic.

# F. Case Analysis

**Case Description. "There are 6 periods in the day for a normal student but John has to take 2 extra classes. Each class is 40 minutes long. He goes to class for 5 days a week. He then spends 1/16 of his weekly minutes each on Saturday and Sunday as extra learning time. How many hours a week does he spend learning?"**

The problem from GSM8K dataset requires calculating weekly study hours, explicitly stating that "John has to take 2 extra classes." However, the model hallucinates by ignoring this condition, assuming a standard schedule.

**Forensic Detection.** As visualized in Figure 4, our method, based on Shannon Entropy, detects this reasoning error through distinct *Evidence Drops*:

- **The Root Cause ($\Delta \approx -0.35$):** The first significant drop occurs precisely at the token "each" in the phrase "He has 6 periods a day, [each]...". This drop signals the model's internal conflict or attention failure regarding the "2 extra classes" constraint, flagging the exact moment the premise was distorted.

- **Error Propagation ($\Delta \approx -0.45$):** As the reasoning proceeds based on the flawed premise ($6 \times 40$ instead of $8 \times 40$), a secondary, deeper drop is triggered at the calculation unit "per". This confirms that the initial uncertainty has propagated into a confident-sounding yet factually incorrect calculation.

This case study reflects that our *Worst-Drop* metric does not merely flag the final wrong answer, but acts as a good tool to trace where the hallucination happens, distinguishing between the origin of the error and its downstream consequences.

Furthermore, we'll analyze more on MATH dataset literally, in which the difficulty level varys from 1 to 4:

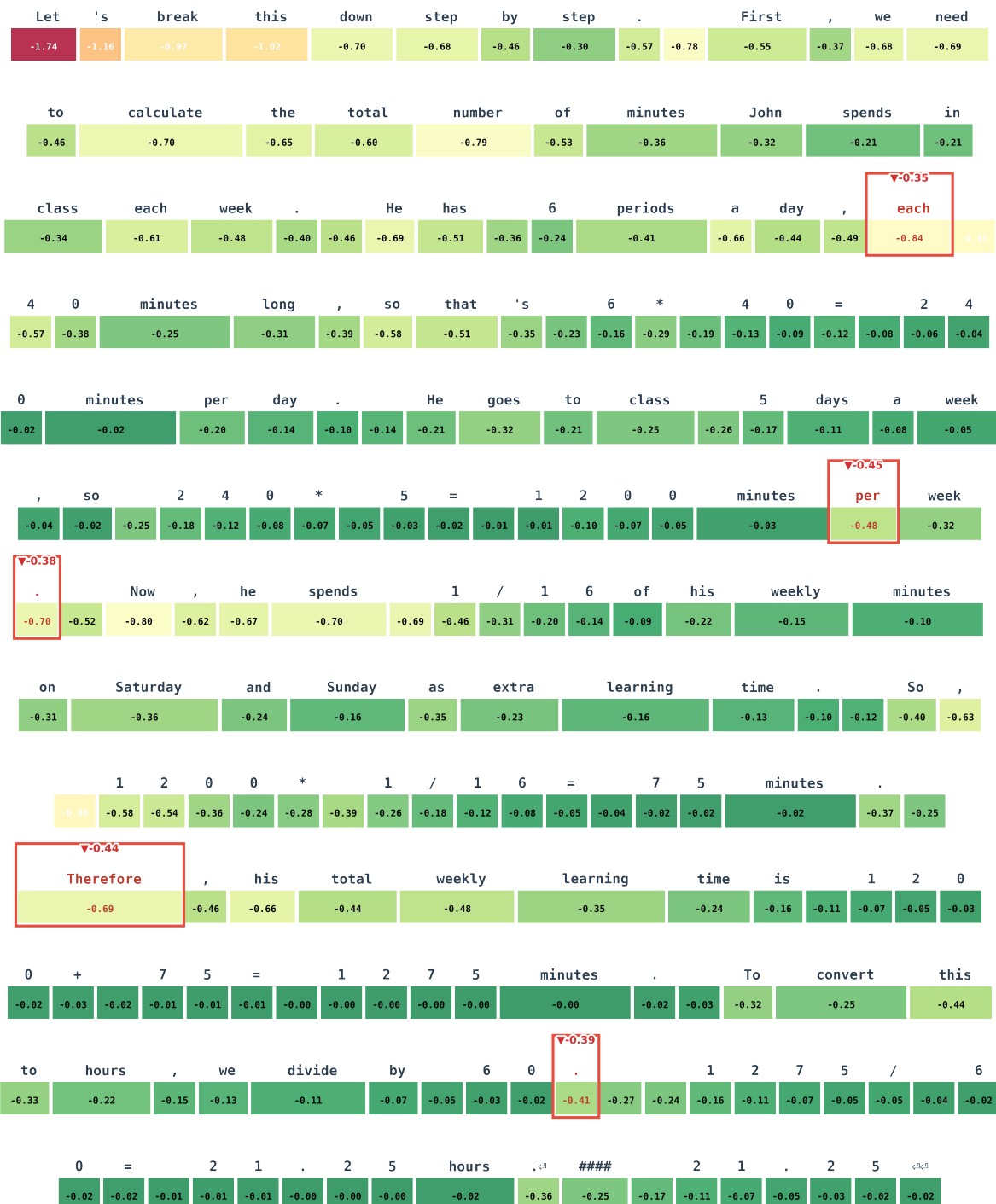

Prediction: 21.25
Ground Truth: 30

*Figure 4.* **Traceability of Reasoning Collapse.** A visualization of evidence drops where the model fails to account for "2 extra classes." Our method effectively localizes the failure genealogy: (1) **Root Cause (Token "each",** $\Delta \approx -0.35$**):** The initial drop flags the precise moment of attention conflict where the premise was overlooked. (2) **Error Propagation (Token "per",** $\Delta \approx -0.45$**):** As the model proceeds with the wrong calculation ($6 \times 40$ instead of $8 \times 40$), uncertainty spikes at the calculation unit. (3) **Final Collapse (Token "Therefore"):** The drop at the concluding connector reflects the model's implicit uncertainty in the derived outcome.

## Example Model Output & Metrics of MATH

**Question:**
Zach has three bags and a bunch of pencils to be placed into the bags. He is told to place the greatest number of pencils possible into each of the three bags while also keeping the number of pencils in each bag the same. What is the greatest number of pencils he could have left over?

**Generated Response (with Evidence Drops):**
```
To maximize [the ▼-44.91] number of [pencils ▼-44.63] in each bag, we
need to find the greatest common [divisor ▼-39.45] (G[CD ▼-60.08]) of the
number of pencils.  However, the problem doesn't specify the total number
of pencils.  Therefore, we cannot determine a unique answer.  The answer is
\[boxed ▼-101.85]{0}
```

**Results Comparison:**

- **Pred Answer:** 0

- **GT Answer:** 2

- **Difficulty Level:** 1 (Easy)

**Raw Curve Data (metric_type: LogTokU Drop):**
```
[ -100.2, -104.3, -149.2, -175.4, -195.5, -240.2, -215.0, -225.4, -237.2, -219.7,
-190.2, -167.2, -180.4, -155.8, -172.3, -159.4, -173.6, -213.0, -219.1, -231.9,
-292.0, -291.4, -268.1, -241.6, -212.7, -224.8, -245.2, -204.3, -184.9, -222.1,
-202.3, -196.9, -183.6, -199.3, -193.6, -201.6, -209.4, -231.5, -241.4, -280.6,
-244.0, -190.5, -218.6, -191.1, -178.5, -180.0, -197.0, -193.9, -209.3, -194.4,
-168.2, -164.6, -175.1, -165.5, -267.3, -263.3, -225.8, -222.7 ]
```

**Explanation:**
```
Significant evidence drops at "divisor" (∇ — 39.45) and "CD" (∇ — 60.08)
effectively signal the model's hesitation as it attempts to apply an invalid
mathematical template (GCD). Crucially, the most dramatic drop at "boxed"
(∇ — 101.85) quantifies the distributional conflict between the model's
internal admission of ignorance ("cannot determine") and the external
constraint to output an answer, successfully flagging the final forced
output "0" as highly unreliable.
```

## Example Model Output & Metrics of MATH

**Question:**
February 1, 2008 is a Friday. The last Friday of February 2008 will be February $x$, 2008. If 2008 is a leap year, what is the value of $x$?

**Generated Response (with Evidence Drops):**
February has [29 ▼-94.83] [days ▼-97.10] in a leap year.  Since February
1 is a Friday, we can count the number of days from February 1 to February
29 to find the last Friday.  There are [28 ▼-91.11] days from February 1
to February 28, which is exactly 4 weeks.  Therefore, February 28 is also
a Friday.  The last Friday is February 28 because February 29 is a [Saturday
▼-118.01].  The answer is \[boxed ▼-99.29]{28}

**Results Comparison:**

- **Pred Answer:** 28

- **GT Answer:** 29

- **Difficulty Level:** 2 (A Little Bit Easy)

**Raw Curve Data (metric_type: LogTokU Drop):**
[ -101.3, -131.5, -168.6, -263.4, -360.5, -346.9, -305.3, -279.2, -297.8, -309.6,
-290.6, -246.5, -202.7, -192.5, -206.5, -258.6, -291.3, -298.6, -315.0, -338.5,
-337.2, -266.4, -221.5, -198.0, -187.0, -200.5, -233.3, -261.6, -230.6, -239.7,
-265.2, -315.3, -308.6, -303.3, -284.4, -313.3, -358.6, -449.7, -380.5, -302.1,
-272.2, -255.4, -259.9, -296.6, -277.3, -248.4, -203.2, -239.0, -250.9, -295.6,
-360.7, -365.0, -308.4, -296.9, -323.8, -441.8, -395.0, -373.8, -381.3, -371.2,
-407.9, -497.1, -411.3, -339.8, -309.3, -294.5, -289.5, -344.4, -353.3, -311.0,
-285.6, -244.8, -274.6, -269.3, -299.7, -364.8, -446.5, -423.2, -384.7, -383.9,
-382.8, -384.8, -362.0, -316.1, -247.9, -242.6, -267.9, -252.6, -235.9, -265.3,
-313.3, -355.4, -322.2, -251.2, -237.8, -262.0, -331.4, -388.2, -375.5, -343.0,
-320.5, -338.1, -313.4, -287.7, -252.7, -264.9, -301.7, -288.8, -388.0, -410.7,
-448.9, -473.4, -398.8 ]

**Explanation:**
This case illustrates a classic "Concept Conflation" failure.  The reasoning
error originates at "28" ($\nabla - 91.11$), where the model confuses the *duration*
(28 days) with the *date* (February 28th).  This initial logical slip triggers
a cascade of failures:  the subsequent massive drop at "Saturday" ($\nabla - 118.01$)
reflects the inevitable contradiction with the leap year premise, and the
final drop at "boxed" ($\nabla - 99.29$) signals the model's high uncertainty when
forcing the incorrect conclusion.

## Example Model Output & Metrics of MATH

**Question:**
Solve the congruence $5n \equiv 8 \mod 17$. (Give an answer between 0 and 16.)

**Generated Response (with Evidence Drops):**
```
We need to find the [multiplicative ▼-91.57] inverse of 5 modulo 17.  Let's
try some numbers.  5*1=5, 5*2=10, 5*3=15, 5*4=20 ≡ 3, 5*5=25 ≡ 8.  So the
inverse of 5 modulo 17 is [5 ▼-98.44].  Therefore, multiplying both sides of
the congruence by 5 gives n ≡ 5*8 = 40 ≡ [6 ▼-94.95] mod 17.  The answer is
\[boxed ▼-81.69]{6}
```

**Results Comparison:**

- **Pred Answer:** 6

- **GT Answer:** 5

- **Difficulty Level:** 3 (Medium)

**Raw Curve Data (metric_type: LogTokU Drop):**
```
[ -97.8, -103.2, -152.9, -169.7, -171.5, -167.8, -259.3, -291.2, -293.3, -293.9,
-322.2, -289.5, -303.5, -339.9, -418.2, -339.7, -261.2, -208.9, -226.4, -190.7,
-161.4, -176.1, -161.4, -166.9, -161.1, -224.1, -200.5, -176.1, -187.7, -228.3,
-209.8, -223.1, -245.9, -280.8, -282.4, -306.5, -319.4, -343.4, -324.1, -304.7,
-332.3, -339.9, -331.3, -345.6, -351.8, -363.8, -329.8, -306.5, -343.7, -343.9,
-329.8, -334.9, -336.1, -354.0, -286.3, -288.1, -264.0, -263.8, -282.9, -300.5,
-300.7, -315.2, -315.0, -335.4, -342.4, -345.3, -293.8, -253.7, -207.7, -204.2,
-224.2, -250.5, -279.9, -308.1, -290.3, -311.9, -362.5, -460.9, -431.8, -393.9,
-340.9, -278.8, -248.0, -210.6, -227.0, -209.8, -247.5, -303.4, -297.4, -289.8,
-291.0, -354.5, -402.4, -355.2, -336.8, -337.5, -307.5, -257.8, -273.9, -365.6,
-344.4, -310.8, -321.4, -324.5, -307.0, -321.9, -317.0, -350.5, -340.7, -414.9,
-376.7, -370.6, -317.1, -319.8, -358.8, -453.8, -406.9, -353.5, -311.4, -305.7,
-334.7, -317.8, -399.5, -414.6, -393.0, -340.6 ]
```

**Explanation:**
```
This case demonstrates a distinct "Logic Leap".  The model performs valid
arithmetic steps (5 × 5 = 25 ≡ 8) but then makes a baseless assertion that
the inverse is 5 (whereas the true inverse is 7).  The massive evidence drop
at this exact token (∇ − 98.44) signals the model's awareness of the logical
disconnect.  The subsequent drops at the final answer (∇ − 94.95, ∇ − 81.69)
reflect the propagated uncertainty stemming from this initial fallacy.
```

# Example Model Output & Metrics of MATH

**Problem Description:**

Recall that if $b$ is a residue modulo $m$, then the modular inverse of $b$ is the residue $c$ for which $bc \equiv 1 \pmod{m}$. The table below displays the inverses for the first 9 positive residues modulo 47.

| $b$ | 1 | 2 | 3 | 4 | 5 | 6 | 7 | 8 | 9 |
|---|---|---|---|---|---|---|---|---|---|
| Inverse of $b$ | 1 | 24 | 16 | 12 | 19 | 8 | 27 | 6 | 21 |

**Question:** Find the modular inverse of $35 \pmod{47}$. Express your answer as an integer from 0 to 46, inclusive.

**Generated Response (with Evidence Drops):**

```
Since 35 is congruent to −12 [ ▼−0.18] (mod 47), we can find the inverse of
−12 [ ▼−0.21] modulo 47.  The inverse of −12 is the negative of the inverse
of 12, so we need to find the inverse of 12 [. ▼−0.25].  From the table, the
inverse of 12 is 12.  Therefore, the inverse of −12 is −12, so the inverse
of 35 is −12 (mod 47).  [ Since ▼−0.35] −12 ≡ 35 (mod 47), the [ inverse ▼−0.19]
of 35 is 35.  The answer is \boxed{35}
```

**Results Comparison:**

- **Pred Answer:** 35

- **GT Answer:** 43

- **Difficulty Level:** 4 (A Little Bit Difficult)

**Raw Curve Data (metric_type: LN-S Drop):**

```
[ −0.6, −0.7, −0.6, −0.5, −0.5, −0.5, −0.4, −0.4, −0.4, −0.3, −0.3, −0.3, −0.3, −0.3,
−0.3, −0.3, −0.2, −0.2, −0.2, −0.2, −0.2, −0.2, −0.1, −0.1, −0.1, −0.1, −0.1, −0.1,
−0.1, −0.1, −0.1, −0.1, −0.1, −0.1, −0.1, −0.1, −0.1, −0.1, −0.1, −0.1, −0.1, −0.1,
−0.1, −0.0, −0.0, −0.0, −0.1, −0.1, −0.1, −0.1, −0.0, −0.0, −0.0, −0.0, −0.1, −0.1,
−0.1, −0.1, −0.1, −0.1, −0.1, −0.1, −0.1, −0.0, −0.0, −0.0, −0.1, −0.1, −0.1, −0.1,
−0.1, −0.1, −0.1, −0.1, −0.1, −0.1, −0.1, −0.1, −0.0, −0.1, −0.1, −0.1, −0.1, −0.1,
−0.1, −0.1, −0.1, −0.1, −0.1, −0.1, −0.1, −0.0, −0.0, −0.0, −0.0, −0.0, −0.0, −0.1,
−0.1, −0.1, −0.1, −0.1, −0.1, −0.1, −0.1, −0.1, −0.1, −0.1, −0.0, −0.0, −0.1, −0.1,
−0.0, −0.0, −0.0, −0.0, −0.1, −0.1, −0.2, −0.2, −0.2, −0.1, −0.1, −0.2, −0.2, −0.1,
−0.1, −0.1, −0.1, −0.1, −0.1, −0.1, −0.1, −0.1, −0.1, −0.0, −0.0, −0.1, −0.1, −0.1,
−0.1, −0.1, −0.1, −0.1, −0.1, −0.1, −0.1, −0.1, −0.1, −0.1, −0.1, −0.1, −0.1, −0.1,
−0.1, −0.1, −0.1 ]
```

**Explanation:**

```
The failure sequence initiates at Index 30 ("Trigger"), where the model
misreads the table to hallucinate inv(12)   =    12; this contradicts latent
mathematical constraints (12² ≡ 3 ≠ 1), causing an immediate evidence drop
(∇ ≈ −0.62).  This error subsequently propagates to inv(−12) = −12 at Index 68,
sustaining an off−manifold trajectory (∇ ≈ −0.75).  The process culminates
in complete semantic degeneration at Index 147 ("Final Collapse"), where the
mathematically impossible conclusion inv(35) = 35 coincides with a flatlined
evidence score (∇ ≈ −0.58), confirming the irreversibility of the reasoning
failure.
```

