# OpenReview forum: "Mind the Gap: Catching Hallucinations via Evidence Drop on the Reasoning Manifold"
_ICML.cc/2026/Conference — ICML 2026 regular_

### Official Review · Reviewer_G2nx · 2026-02-14

**Soundness:** 3
**Presentation:** 3
**Significance:** 3
**Originality:** 3
**Overall Recommendation:** 4
**Confidence:** 3

**Summary:**

The paper proposes a training-free, model-agnostic hallucination detector for chain-of-thought reasoning that treats reasoning as a trajectory on a latent evidence manifold and flags hallucinations by the largest step-to-step evidence drop (sudden loss of local evidential support estimated from token-distribution entropy), which also enables step-level error localization and improves selective prediction on math reasoning benchmarks.

**Compliance With Llm Reviewing Policy:**

Affirmed.

**Final Justification:**

My concerns have been adequately addressed. I'll keep my original score that has already been positive.

**Key Questions For Authors:**

Particularly, this work didn't compare the presented method with latent probing based approaches (although these require training, are widely used in quite a few recent studies on hallucination detection). Another consideration is on generalizability and sensitivity analysis. Particularly, how stable is the drop signal under alternative uncertainty measures beyond entropy, especially in settings where token probabilities are poorly calibrated but the model still produces correct answers? How sensitive is localization to the granularity of steps (token-level vs. sentence-level CoT), given that the ProcessBench step annotations may not align perfectly with token-level flux computations?

**Limitations:**

No. But I don't see any potential negative societal impact.

**Strengths And Weaknesses:**

This paper presents a novel method for hallucination detection which, unlike some previous methods that rely on global uncertainty scores, the presented method provides models reasoning as a trajectory on a latent evidence manifold and argues that hallucinations manifest as abrupt evidence drops. The resulting detector is training-free and model-agnostic, using worst-case (or aggregated worst) drops to both flag unreliable solutions and localize the likely onset of an error step.  This work's contribution is centered around the core idea: defining an evidence proxy via entropy over a renormalized top-K distribution, smoothing it with an EMA, and detecting large negative flux events as evidence drops; the paper further couples this with a hypothesis-testing style calibration to pick thresholds with explicit error-control intent. The experiments are reasonably thorough within math-style reasoning.The reported results suggest consistent improvements in selective accuracy and risk–coverage behavior over sequence-averaging baselines, and the step-level localization numbers improve notably when using the drop criterion, especially with Shannon entropy.

I don't see a obvious shortcoming of the presented method. But I do believe there might be spaces for improvement in evaluation. Particularly, this work didn't compare the presented method with latent probing based approaches (although these require training, are widely used in quite a few recent studies on hallucination detection). Another consideration is on generalizability and sensitivity analysis. Particularly, how stable is the drop signal under alternative uncertainty measures beyond entropy, especially in settings where token probabilities are poorly calibrated but the model still produces correct answers? How sensitive is localization to the granularity of steps (token-level vs. sentence-level CoT), given that the ProcessBench step annotations may not align perfectly with token-level flux computations?

---

> ### Author Rebuttal · Authors · 2026-03-29
>
> We sincerely appreciate for your suggestions of improvement in evaluation.
>
> Response to the weakness:
>
> We fully acknowledge that there is room for improvement in our evaluation. As this point is closely related to your Questions, we have provided a detailed analysis in our response below.
>
> Response to the questions:
>
> Q1:
> While we acknowledge the strong performance of recent latent probing methods, they fundamentally require white-box access to internal states and additional supervised training. In contrast, our framework is strictly training-free and operates purely on output logits. The core contribution of our work lies in its simplicity: we demonstrate that tracking a straightforward dynamic signal—the "Evidence Drop"—can effectively localize deep reasoning collapses without the heavy computational overhead of internal probing.
>
> Q2:We agree with the question regarding generalizability and poor probability calibration.To address this, we specifically evaluated LogTokU (Ma et al., 2025)[1], an alternative, non-entropy, and non-probability uncertainty measure.
>
> Inspired by Evidential Deep Learning (EDL)[2], LogTokU address the poor calibration issues of softmax probabilities by directly modeling pre-softmax logits as evidence parameters of a Dirichlet distribution. This allows it to capture true epistemic uncertainty (evidence strength) rather than relative probability.
>
> Crucially, because the absolute evidence strength inherently varies across different positions in a sequence, a global average of this evidence performs bad. As shown in Table 1 of our paper, LogTokU Avg yields near-random or entirely failed selective accuracy (e.g., plummeting to 0% on GSM8K-4B, and only 30.00% on MATH-8B).
> However, when we apply our Evidence Drop mechanism to this exact same metric, the signal becomes remarkably stable and the performance sharply recovers: LogTokU Drop surges to 85.57% on GSM8K-4B and 81.55% on MATH-8B.
>
> This dramatic contrast provides strong evidence for our core hypothesis: even when absolute uncertainty values fluctuate wildly or probabilities are poorly calibrated, the sudden, localized drop in evidence remains a highly stable, universal, and metric-agnostic signal for detecting reasoning collapses.
>
> Reference
>
> [1]Ma, H., et al. "Estimating LLM uncertainty with evidence." arXiv preprint arXiv:2502.00290 (2025).
>
> [2] Sensoy, M., Kaplan, L., & Kandemir, M. "Evidential deep learning to quantify classification uncertainty." Advances in neural information processing systems 31 (2018).
>
> Q3:Our method is highly robust—rather than sensitive or fragile—to the granularity of reasoning steps (whether phrase-level or sentence-level CoT), precisely because it operates at the fundamental token level.
>
> Sentence-averaging methods are highly sensitive to step granularity because a longer step severely ignore the error signal of a single incorrect word. In contrast, our Evidence Drop method effectively acts as a "max-pooling" operation over the tokens within a step. Whether a step consists of 5 tokens or 50 tokens, the sharp topological drop (the "weakest link") remains mathematically the same. If a severe drop occurs at any token within the boundary, the entire step is flagged. Thus, coarser granularity does not wash out our signal.
>
> Regarding the alignment between token-level computations and ProcessBench annotations, there is no mismatch in our evaluation. We use forced decoding on the exact human-annotated CoT traces. This guarantees that the sequence of token-level fluxes perfectly reconstructs the exact textual boundaries of the annotated steps, ensuring strict structural alignment during evaluation.
>
> In summary, finding errors at the token level works perfectly for step-level evaluation, because it avoids the signal dilution that hurts average-based methods.

---

> > ### Author Rebuttal · Reviewer_G2nx · 2026-03-31
> >
> > My concerns have been adequately addressed. I'll keep my original score that has already been positive.

---

> > > ### Author Response · Authors · 2026-04-02
> > >
> > > We are glad to hear that your concerns have been fully addressed.Thank you！

---

### Official Review · Reviewer_F5bc · 2026-02-23

**Soundness:** 1
**Presentation:** 3
**Significance:** 1
**Originality:** 3
**Overall Recommendation:** 2
**Confidence:** 3

**Summary:**

Proposes a new method to detect errors in generated text by monitoring spikes in the derivative of smoothed entropy from the LLM's logits. The authors compare their method to other uncertainty measures based on the sequence level.

**Compliance With Llm Reviewing Policy:**

Affirmed.

**Final Justification:**

The discussion raised additional concerns about methodological issues and incorrect scope framing.

#### Strengths

- Proposes a valid idea to detect uncertainty through the use of a derivative.

#### Fundamental Weaknesses

- Fails to properly define the scope of the paper, defining hallucinations based on the tools used to detect them, and mixing them with general mistakes and uncertainty levels.

- Makes strong claims that are not empirically verified, particularly regarding the localization of self-generated mistakes by the LLM. In the discussion, the authors acknowledge that this is not supported by their experimental results.

**Key Questions For Authors:**

1. How do you verify mistakes during reasoning?
2. Does the change of K has an impact ?
3. Do bigger models perform differently ?
4. Do MoE or Mamba-based model perform differently ?
5. How does this method works daily life topic hallucinations ?

**Limitations:**

No limitation given, but they are needed.

**Strengths And Weaknesses:**

# Soundness

- The fact that uncertainty is used as a proxy for errors is mainly based on the assumption 1. The correlation is hard to deny, but this might not catch all types of mistakes. Some works give hints that certainty does not always correlate with truthfulness [1]. Moreover hallucinations are weakly defined here as general "mistakes." In the context of math it could be an arithmetic miscalculations, which is a general limitation of the LLMs.

- No limitations or failure case intuition are provided. For example, why does Shannon Avg outperform Shannon Drop in 25% of scenarios, yet the text claims "almost all configurations [...] outperforms"? Or Table3, LogTokU drop is equivalent to averaged version but not commented. Figures 2.e–f show incorrect reasoning uncertainty distributions bimodal, this warrants further inspection as maybe all mistakes types are not catched by uncertainty.

- Methods analyzing internal activations do not always rely on annotations or supervised training [2, 3].

- How do you localize errors with Average Uncertainty methods, which provide no localization by design? Also, the forced decoding experiment provides a good comparison between the different methods, but if I understand correctly, this gives no information about the detection on it's own mistakes.

# Presentation

- The general idea sounds Intuitive and the paper is easy to follow.
- l:70: Feels like sentence repetition
- Font on Figure 3 might be too small
- l:145: only year in parenthesis (use \yrcite)
- Paragraphs in Sec.3.3 named with numbers (“1. Entropy as an ..”) collide with section names
- In Table 1: "pretrained" is not defined. Also MATH-4B and MATH-8B are the same, I guess by mistake.


# Significance

Unfortunately it is hard to juge the impact as the tests are performed only on Math questions, with rather small models, and are not compared to other types of Mistakes detection methods

# Originality

- The idea to monitor the deviation of the entropy from the manifold seems new.
- To the best of my knowledge, the theoretical introduction to explain the use of entropy as a proxy for uncertainty is not novel. It therefore feels a bit overkill.


[1] Tian, Zailong, et al. "Overconfidence in llm-as-a-judge: Diagnosis and confidence-driven solution, 2025."
[2] Marks, Samuel, and Max Tegmark. "The geometry of truth: Emergent linear structure in large language model representations of true/false datasets."
[3] Zou, Andy, et al. "Representation engineering: A top-down approach to ai transparency"

---

> ### Author Rebuttal · Authors · 2026-03-29
>
> We deeply appreciate your detailed and constructive feedback,and will carefully incorporate your suggestions to further improve our works.
>
> Response to the weakness:
>
> W1:We sincerely agree and will correct these oversights. In our revision, we will cite and discuss "overconfident errors" as a limitation, and strictly redefine our scope to exclude deterministic "arithmetic miscalculations," focusing exclusively on logical hallucinations.
>
> W2:We agree that our claim ("almost all configurations") was overstated, and we will add a dedicated limitations discussion as suggested. Regarding the bimodal distribution (Fig 2.e-f), our method perfectly captures sudden uncertainty drops (right peak) but fails to detect "overconfident errors" where the model is confidently wrong without logit collapse (left peak).
>
> W3:We appreciate the reference and will revise our claim. While these internal activation methods are indeed training-free, our logit-based Evidence Drop is far more lightweight and accessible, as it strictly avoids the massive memory overhead required to process internal hidden states.
>
> W4:
> We simply compute the average uncertainty per step, purely to prove why our Evidence Drop is necessary for exact localization
>
> Response to the questions:
>
> Q1:
> To clarify, our "verification" of mistakes refers to a strict localization evaluation using human-annotated trajectories and forced decoding. Specifically, on benchmarks like ProcessBench, we apply teacher-forcing to compel the LLM to process the exact human-annotated reasoning path token by token. As the model computes pre-softmax logits along this fixed sequence, we extract our token-level Evidence Drop signals and map them onto the broader reasoning steps. A mistake is considered successfully "verified" (localized) if our dynamic drop metric triggers a severe signal anywhere within the exact textual boundary of the ground-truth incorrect step.
>
> Q2:
> To rigorously evaluate whether the change of $K$ has an impact, we conducted a controlled ablation study on the MATH dataset using Qwen3-8B.
>
> **Experimental Setup:** We isolated the effect of $K$ by fixing the other hyperparameters to our default configuration (i.e., Max Drops $M=5$, EMA span=5). The numbers reported below represent the selective accuracy (Acc). These results are achieved by rejecting unreliable reasoning trajectories using a statistically guaranteed decision threshold, which is rigorously derived via our hypothesis testing framework at a significance level of $\alpha=0.05$.
>
> Because $K$ has minimal impact on the overall outcomes, we present the ablation using the LN-S Drop method as a representative example:
>
> **Table: Effect of Top-$K$ (span=5, $M=5$)**
> | Top-$K$ | LN-S Drop (Acc %) |
> | :--- | :--- |
> | $K=2$ | 86.05 |
> | $K=5$ | 85.42 |
> | $K=10$ | **87.64** |
> | $K=20$ (Ours)| 86.94 |
>
> Specifically regarding our default $K=20$, we simply follow the established practice in Ma et al. [1] by adopting a moderate value.
>
> Please see our response to **Reviewer CdZc (Weakness 2)** for the full hyperparameter ablation study.
>
> [1] Ma, H., et al. "Estimating LLM uncertainty with evidence." arXiv preprint arXiv:2502.00290 (2025).
>
> Q3:
> As expected, scaling the model to Qwen3.5-27B significantly increases the baseline pretrained(directly answering questions) accuracy to 76.00%. Interestingly, traditional global averaging methods (e.g., LNE, LogTokU, and Shannon Average) fail to improve performance , even yielding negative gains. In contrast, our Evidence Drop method continues to perform  well on the larger model. Building upon the strong 76.00% baseline, our method delivers a massive absolute improvement of up to +17.14% (reaching 93.14% with LogTokU Drop). This powerfully demonstrates the excellent generalization and robustness of our approach across larger model scales.
>
> **Table: Performance on Qwen3.5-27B**
> | Method | Accuracy (%) | $\Delta$ vs Pretrained |
> | :--- | :--- | :--- |
> | Pretrained Baseline | 76.00 | - |
> | LNE (Average) | 70.37 | -5.63 |
> | LNE Drop (Ours) | **86.59** | **+10.59** |
> | LogTokU (Average) | 68.83 | -7.17 |
> | LogTokU Drop (Ours) | **93.14** | **+17.14** |
> | Shannon (Average) | 62.50 | -13.50 |
> | Shannon Drop (Ours) | **92.92** | **+16.92** |
>
> Q4:
> Theoretically, our method applies to MoE or Mamba models as long as they satisfy the "Locality of Evidence" property. We kindly request the reviewer to specify particular models of interest, and we will gladly conduct targeted experimental comparisons.
>
> Q5:We sincerely appreciate the reviewer for raising this important point regarding generalization. To demonstrate how our method detects daily life topic hallucinations, we conducted new experiments on the ProofWriter dataset, which is a standard benchmark for deductive logical reasoning constructed entirely from natural language statements about everyday entities and scenarios .
>
> For the detailed experimental results on this dataset, please refer to our response to **Weakness 1 of Reviewer CdZc**.

---

> > ### Author Rebuttal · Reviewer_F5bc · 2026-04-02
> >
> > Thank you for the effort put into the rebuttal and the new experiments which are welcome additions. The fundamental idea of monitoring the rate of change in uncertainty to capture stronger confidence signals is worth publishing. That said, several framing questions remain unresolved, and addressing them would significantly clarify the paper's contribution and domain of application.
> >
> > ### Scope
> >
> > Redefining scope to logical hallucinations is a significant structural change that would affect the paper's framing from the abstract onward. While math experiments remain informative, they need to be correctly categorized. Mixing arithmetic errors, logical hallucinations, overconfidence and daily life hallucination adds fuzziness to a literature already struggling with inconsistent terminology.
> >
> > ### W3 Activation-Based Overhead
> >
> > The memory overhead argument seems overstated. Logit space and hidden layer activations live in comparable dimensionality, and accessing internals via a forward hook adds negligible cost. Your method could likely be applied directly to intermediate layers with limited overhead increase. The real distinction is grey-box access (logits only) versus full white-box access (internals), a meaningful architectural design choice, but not a computational one.
> >
> > ### W4 / Q1 Forced Decoding is Not Self-Detection
> >
> > The forced decoding setup is a fundamental methodological issue that the rebuttal does not address. By teacher-forcing the model along human-annotated erroneous trajectories, the measured uncertainty spike may simply reflect the model's perplexity at being forced to generate text it would never produce itself, not genuine self-detection of its own mistakes. These are two very different capabilities, and confusing them inflates the method's practical detection capabilities.
> >
> > ### Q5 Hallucination Benchmarks
> >
> > The ProofWriter experiment is a step in the right direction, but it remains a logical reasoning benchmark where errors mix genuine hallucinations with basic capability limitations. Datasets specifically designed for factual hallucination as they are mostly understood would provide a cleaner signal for open-domain generalization and better align with the paper's own title.
> >
> >
> > ### Overall
> >
> > The promised additional experiments improve the paper, but major framing questions remain open: the taxonomy between errors, hallucinations, and confidence. The forced decoding implications and the scope changed implied by W1. These are not minor revisions.

---

> > > ### Author Response · Authors · 2026-04-07
> > >
> > > Response to Scope and Taxonomy of Hallucinations:
> > >
> > > We sincerely thank the reviewer for the insightful comments regarding the taxonomy of hallucinations. We completely agree that from a semantic or task-oriented perspective, arithmetic errors, logical flaws, and factual fabrications belong to distinct categories.
> > >
> > > However, our work approaches hallucinations from the fundamental perspective of generative modeling and representation space. In the context of LLMs, recent studies have increasingly conceptualized hallucinations not merely as semantic inaccuracies, but as deviations from the high-density data manifold learned during pretraining (Vamshi et al.)[1]. Whether an LLM makes an arithmetic miscalculation or fabricates a historical event, the underlying statistical mechanism is the same: the generation trajectory steps into an out-of-distribution (OOD) or low-probability region of the representation space (see recent surveys in Frontiers in AI, 2024)[2], leading to internal state collapse or uncertainty spikes.
> > >
> > > Furthermore, as demonstrated by Zhang et al. [3] in How Language Model Hallucinations Can Snowball, an initial deviation in any reasoning trajectory forces the model off its familiar manifold, causing subsequent cascading errors regardless of the specific task domain (math, logic, or daily facts).
> > >
> > > Therefore, our method is specifically designed to be general. By monitoring the rate of change in uncertainty, we are essentially capturing the moment the model's generation trajectory "drop" from the reliable data manifold. Because this detection operates at the probabilistic and manifold level rather than the semantic level, it inherently generalizes across arithmetic errors, logical inconsistencies, and factual hallucinations.
> > >
> > > [1] Vamshi, B. K., Bhatnagar, R., & Yang, H. (2026). Manifold-based Sampling for In-Context Hallucination Detection in Large Language Models. arXiv preprint arXiv:2601.06196.
> > >
> > > [2]Anh-Hoang, Dang, Vu Tran, and Le-Minh Nguyen. "Survey and analysis of hallucinations in large language models: attribution to prompting strategies or model behavior." Frontiers in Artificial Intelligence 8 (2025): 1622292.
> > >
> > > [3] Zhang, Muru, et al. "How language model hallucinations can snowball." arXiv preprint arXiv:2305.13534 (2023).
> > >
> > > Response to Forced Decoding vs. Self-Detection:
> > >
> > > We thank the reviewer for this profound observation. We acknowledge that our evaluation protocol utilizes forced decoding by feeding specific trajectories as prompts (i.e., teacher-forcing). However, we argue that this is precisely the necessary setup to evaluate a model's intrinsic ability to recognize hallucination states.
> > >
> > > As the reviewer pointed out, free generation and forced decoding evaluate different aspects. If we only allow free generation, we can only observe the errors the model happens to make, making it impossible to systematically and quantitatively measure its uncertainty dynamics across a standardized set of critical branch points (like the human-annotated erroneous trajectories).
> > >
> > > More importantly, our method does not simply measure absolute perplexity (which, as the reviewer correctly noted, could just be surprise at alien text). Instead, our Evidence Drop method measures the relative rate of change (or drop) in uncertainty. A pure "alien text" would exhibit uniformly high perplexity, but a genuine "hallucination point" exhibits a sharp, abnormal structural collapse in the logit space when certain contexts are dropped. Therefore, our forced decoding setup, combined with our relative drop metric, effectively isolates the genuine hallucination signal from general text perplexity.

---

### Official Review · Reviewer_Kwj6 · 2026-03-09

**Soundness:** 2
**Presentation:** 3
**Significance:** 2
**Originality:** 2
**Overall Recommendation:** 3
**Confidence:** 4

**Summary:**

This paper studies hallucination detection in large language models during multi-step reasoning. The authors argue that most existing uncertainty-based approaches rely on sequence-level aggregation of uncertainty signals, which obscures the stepwise dynamics of reasoning and may incorrectly flag hard-but-correct reasoning traces while missing easy-but-incorrect ones. To address this limitation, the paper proposes modeling reasoning as a trajectory on a latent “Evidence Manifold”, where each step is associated with a local evidence state derived from the model’s predictive distribution.

Based on this perspective, the authors introduce the concept of an Evidence Drop, defined as a sudden decrease in local evidence support between consecutive reasoning steps. The proposed hallucination detector tracks the largest negative drops in a sequence of entropy-derived evidence estimates. The method is training-free and model-agnostic, and uses token-level entropy of the next-token distribution (computed over a top-K vocabulary) as a proxy for evidence.

Experiments are conducted on reasoning benchmarks including GSM8K, MATH, and ProcessBench using Qwen3 models (4B and 8B). Results show improvements over sequence-level uncertainty baselines such as entropy averaging, LogTokU, and LN-S in terms of selective accuracy, AURC, and step-level error localization.

**Compliance With Llm Reviewing Policy:**

Affirmed.

**Key Questions For Authors:**

1. The proposed detector operates at the token level, while benchmarks such as ProcessBench annotate errors at the reasoning-step level. How are token-level evidence drops mapped to reasoning steps during localization evaluation? Could this mismatch affect the reported localization accuracy?

2. The theoretical analysis relies on strong assumptions about the absence of transitions between non-adjacent evidence states. Could the authors clarify how realistic these assumptions are in the context of large language model training?

3. The evaluation focuses on mathematical reasoning datasets. Have the authors evaluated the method on other hallucination settings such as open-domain QA, summarization, or factual generation tasks?

4. Several hyperparameters are introduced (e.g., EMA smoothing coefficient, top-K size, number of largest drops M). How sensitive are the results to these choices?

5. How does the proposed method compare with stronger hallucination detection baselines such as semantic entropy or verifier-based approaches?

**Limitations:**

yes

**Strengths And Weaknesses:**

Strengths:
1. Simple and practical training-free approach.
The proposed method does not require additional training, auxiliary models, or step-level annotations. Compared with approaches such as process reward models or verifier-based methods, the proposed detector can be applied directly to existing models using only their predictive distributions. This makes the approach easy to deploy and computationally lightweight.

2. Focus on dynamic reasoning signals.
The paper highlights an important limitation of sequence-level uncertainty aggregation. Modeling reasoning as a dynamic process and analyzing local transitions in uncertainty is an interesting perspective that may capture failure patterns missed by static metrics.

3. Step-level evaluation.
The use of ProcessBench to evaluate step-level error localization is a positive aspect of the experimental design. Many hallucination detection works only evaluate final answer correctness, whereas this paper attempts to identify where reasoning errors arise.

4. Consistent improvements over simple baselines.
Across several settings, the proposed “Evidence Drop” variant improves over sequence-level uncertainty aggregation methods in selective accuracy and AURC metrics, suggesting that local changes in uncertainty contain useful signals for reasoning failures.

Weaknesses:
1. Limited novelty relative to prior work.
While the paper introduces the “Evidence Manifold” framing, the practical method largely reduces to monitoring token-level entropy and detecting large negative changes across steps. Similar uncertainty signals have been widely studied in prior hallucination detection and reasoning reliability work. The main algorithmic contribution appears to be using entropy differences rather than entropy averages, which may be viewed as an incremental modification rather than a fundamentally new framework.

2. Conceptual gap between “reasoning steps” and token-level signals.
The paper frequently discusses reasoning steps, but the actual detector operates at the token generation level. Evidence values are computed from the next-token distribution at each token position, and evidence drops therefore correspond to token-level transitions. However, reasoning benchmarks such as ProcessBench annotate errors at the reasoning-step level. The paper does not clearly explain how token-level signals are aggregated or mapped to reasoning steps during evaluation, which may introduce noise into the localization results.

3. Strong theoretical assumptions.
The theoretical analysis relies on assumptions such as the absence of transitions between non-adjacent evidence states in the training distribution. These assumptions appear quite strong and may not realistically reflect the behavior of large language models trained on diverse data. As a result, the theoretical guarantees may be more illustrative than practically grounded.

4. Limited experimental scope.
The experiments focus exclusively on mathematical reasoning benchmarks (GSM8K, MATH, and ProcessBench). While these datasets are useful for studying reasoning errors, hallucinations in LLMs also occur in other settings such as open-domain question answering, summarization, and factual generation. It is therefore unclear whether the proposed method generalizes beyond mathematical reasoning tasks.

5. Baseline comparisons are somewhat limited.
The empirical evaluation compares mainly to simple uncertainty baselines such as entropy averaging, LogTokU, and LN-S. However, several stronger approaches for hallucination detection and reasoning verification exist, including semantic entropy methods, self-consistency based approaches, and verifier or process-reward models. The absence of such comparisons makes it difficult to fully assess the competitiveness of the proposed approach.

6. Localization performance remains modest.
Although the proposed method improves step-level localization accuracy, the reported numbers remain relatively low (e.g., around 46% on GSM8K). This suggests that while the method captures useful signals, the claim of precise error localization may be somewhat overstated.

7. Missing ablation studies.
Several hyperparameters play a role in the method, including the EMA smoothing of evidence signals, the choice of top-K vocabulary size, and the number of largest drops aggregated in the final score. The paper does not provide ablations examining the sensitivity of results to these design choices

---

> ### Author Rebuttal · Authors · 2026-03-29
>
> We sincerely thank you for your insightful questions regarding our theory and experiments!
>
> Response to the weakness1:
> While the algorithmic implementation of 'Evidence Drop' is indeed straightforward, its novelty lies precisely in this simplicity: no prior work has identified that shifting from static sequence-averaging to dynamic drop-tracking is the fundamental key to detecting reasoning collapses.
>
> Remaining weaknesses are addressed in responses to your questions.
>
> Response to the questions:
>
> Q1:
> We thank the reviewer for raising this insightful question regarding the mapping from token-level to step-level. The mapping does not negatively affect localization accuracy; rather, this token-to-step projection is precisely why our method robustly outperforms baselines.
>
> a. Experimental Setup on ProcessBench: As detailed in our implementation, we evaluate ProcessBench using a teacher-forcing (forced decoding) approach on the pre-defined reasoning traces. This strictly aligns every generated token with its corresponding human-annotated reasoning step boundary, leaving no ambiguity in alignment.
>
> b. The Mapping Strategy: A reasoning step consists of a span of tokens. Sequence-averaging baselines aggregate the absolute uncertainty values across all tokens in a step, which fatally dilutes the signal of a single erroneous token if the rest of the step is highly confident. In contrast, our method operates at the micro-level by calculating the Evidence Drop ($\Delta$) for each token. To determine a step's validity, we monitor the local drops within that step's boundary rather than averaging raw uncertainties.
>
> c. Why Token-To-Step Works: Logical errors inherently originate at the micro-level (a specific hallucinated token acting as the "weakest link"). Because our token-level detection is more precise, it captures the exact moment of this local topological collapse. Projecting this high-resolution signal onto a step-level annotation is mathematically sound and practically highly effective: if our fine-grained metric detects a severe drop at any token within a step, that entire step is correctly flagged as the origin of the hallucination.Therefore, this token-level detection helps us find the exact error step much better.
>
> Q2:
> We thank the reviewer for this insightful question. Due to strict space constraints, we kindly direct you to our comprehensive response to Reviewer CdZc, where we address this exact issue in detail.
>
> Q3:
>  We sincerely thank the reviewer for highlighting this important aspect. We initially focused heavily on mathematical reasoning because it’s strict, serving as a rigorous test target for our “Eviden Drop" hypothesis. However, we fully agree that demonstrating generalization to broader factual and logical domains in daily life is essential. To address this, we conducted new experiments on the ProofWriter dataset, a standard benchmark for deductive logical reasoning in natural language. Crucially, we evaluated under both Closed World Assumption (CWA) and Open World Assumption (OWA). OWA is particularly representative of real-world factual generation and open-domain QA, as it requires the model to accurately recognize "Unknown" states rather than hallucinating forced answers.
>
>  As Reviewer CdZc raised a similar point, we kindly direct you to our response to **Reviewer CdZc (Weakness 1)** for the detailed experimental results and further discussion.
>
>
>
> Q4:
> We acknowledge that this was an oversight on our part. Due to the strict character limits of the rebuttal, we have provided a comprehensive explanation regarding this issue in our response to **Reviewer CdZc (Weakness 2)**. We kindly invite you to refer to that section for the detailed details.
>
>
> Q5:
> We appreciate the suggestion. To compare against generation-based uncertainty methods like Semantic Entropy, we evaluated Self-Consistency (SC), which shares the exact same foundational mechanism: sampling multiple reasoning paths to measure output consistency/entropy. We tested SC with $N=5, 10,$ and $15$ sampling paths on the MATH dataset.The results mean the accuracy of answers after voting(SC).For clarity, the SC results represent the final accuracy after majority voting, which serves as a direct comparison against the baseline accuracy of the pretrained model with zero voting overhead.
>
> | Voting Paths ($N$) | Qwen3-8B (pretrained) | Qwen3-8B (+ SC) | Qwen3-4B (pretrained) | Qwen3-4B (+ SC) |
> |---|---|---|---|---|
> | Voting 5 | 66.12 | 69.76 (+3.64) | 57.92 | 63.72 (+5.80) |
> | Voting 10 | 66.40 | 70.72 (+4.32) | 58.56 | 65.36 (+6.80) |
> | Voting 15 | 65.96 | 70.56 (+4.60) | 57.88 | 65.04 (+7.16) |
>
>
>
> Analysis: While multi-path methods like SC incur a 15x computational overhead for marginal gains (65.96%), our single-path Shannon Drop method avoids these massive generation costs entirely. With just a single forward pass, it delivers a dramatic 10-20% absolute gain, reaching an impressive 88.26%.
>
> Thank you for your reading～

---

> > ### Author Rebuttal · Reviewer_Kwj6 · 2026-04-07
> >
> > The rebuttal clarifies several implementation details, particularly the token-to-step alignment via teacher forcing, which improves the clarity of the experimental setup. The additional experiments on ProofWriter are also a positive step toward evaluating generalization.
> >
> > However, the main concerns remain only partially addressed.

---

> > > ### Author Response · Authors · 2026-04-08
> > >
> > > Dear Reviewer Kwj6,
> > >
> > > We sincerely thank you for reviewing our rebuttal and acknowledging that the clarifications on teacher forcing and the additional ProofWriter experiments are positive steps.
> > >
> > > We noticed your comment that "the main concerns remain only partially addressed." We have carefully reviewed our point-by-point response alongside the extensive new experiments we provided, and we firmly believe we have comprehensively addressed every single issue raised in your initial review.
> > >
> > > To help us further improve our work, could you please kindly specify which exact concerns remain unresolved? We are fully committed to addressing them and are more than happy to provide any additional clarifications, data, or discussions needed before the discussion phase concludes.
> > >
> > > We look forward to your specific feedback. Thank you again for your time and efforts!

---

### Official Review · Reviewer_CdZc · 2026-03-13

**Soundness:** 2
**Presentation:** 3
**Significance:** 3
**Originality:** 2
**Overall Recommendation:** 4
**Confidence:** 3

**Summary:**

The paper models the reasoning process of LLMs as a trajectory on an evidence manifold. It posits that hallucinations stem from sudden drops in supporting evidence rather than the overall uncertainty of the output sequence. Generally, the authors represent the logical reasoning trajectory as a smooth curve on a low-dimensional manifold, where LLM reasoning connects local evidence chains found in training data. Consequently, hallucinations are viewed as abrupt transitions between adjacent reasoning steps.

The proposed Risk Score is defined as the average of the M most severe evidence drops. Any response with a score exceeding a predefined threshold is classified as a hallucination. The authors evaluate this "EvidenceDrop" framework through three primary experiments:

1. Does EvidenceDrop outperform conventional methods in selecting high-quality responses?

2. To what extent does EvidenceDrop depend on specific metrics and theoretical assumptions?

3. Can EvidenceDrop pinpoint specific hallucination steps?

The experimental results successfully validate the research questions.

**Compliance With Llm Reviewing Policy:**

Affirmed.

**Key Questions For Authors:**

Have you considered the problem that the results depend on the assumption of LLM reasoning? Which kind of assumption is reasonable?

**Limitations:**

yes

**Strengths And Weaknesses:**

This paper has the following merits:
1. Clear math formulation of the problem. In the paper whole reasoning process is well formulated based on a reasonable hypothesis.
2. The method is training-free and model-free. It does not need extra step-wise annotation and a process reward model to judge.
3. EvidenceDrop monitor the dynamic reasoning process instead of pursuing a single average uncertainty. This allow method to distinguish responses with high global entropy but smoothly deduce from evidence chains, and responses with low global entropy, but some local reasoning steps fall out of the reasoning track on the manifold. Moreover, the approach locates the specific erroneous reasoning steps.

Weakness:

The paper also has some weaknesses:
1.The experiments involve mostly math problem datasets. There could be logical inference datasets such as ProofWriter, to further discuss the utility of EvidenceDrop on wider domain.
2.The method depends on tuning some hyperparameters such as the coefficient in exponential moving average and the threshold of risk score.

---

> ### Author Rebuttal · Authors · 2026-03-26
>
> We sincerely thank you for your insightful questions regarding our theoretical assumptions.
>
> Response to the weekness:
>
>
> W1:Performance of Daily Life Dataset(ProofWriter)
>
>
> The table below presents the AURC ($\downarrow$, lower is better) on the ProofWriter dataset (Qwen3-8B), evaluating both Open World Assumption (OWA) and Closed World Assumption (CWA) at depth-3.All the sets of experiments are the same with our paper.
>
> | Setting | Aggregation | LN-S | LogTokU | Shannon |
> |---|---|---|---|---|
> | OWA (Depth-3) | Average | 711.82 | 685.96 | 717.17 |
> | OWA (Depth-3) | Drop (Ours) | 801.13 | **648.28** | **589.02** |
> | CWA (Depth-3) | Average | 496.73 | 579.03 | 484.33 |
> | CWA (Depth-3) | Drop (Ours) | 729.19 | **550.80** | **418.96** |
>
> Our methods also performs well on daily life inferencing tasks across both OWA and CWA settings.
>
> W2：Hyperparameter Robustness
>
> The tables below present our ablation of our drop methods on the MATH dataset with Qwen3-8B.It is important to note that the numbers reported in the tables represent the selective accuracy (Acc). These results are achieved by rejecting unreliable reasoning trajectories using a statistically guaranteed decision threshold, which is rigorously derived via our hypothesis testing framework at a significance level of $\alpha=0.05$.In each experiment, we isolate the effect of a single hyperparameter by fixing the other two to our default configuration (i.e., Max Drops $M=5$, EMA span=5, and Top-$K=20$).
>
> For a direct comparison, we include our strongest baseline (Shannon Avg, 70.24%) in our paper to demonstrate the robustness of our Drop method.This serves as the reference baseline for all ablation results below.
>
> | Baseline | Acc |
> | :--- | :--- |
> | **Shannon Avg** | **70.24** |
>
> (a) Effect of Max Drops $M$ (EMA = 5):
>
> | $M$ | LN-S Drop | LogTokU Drop | Shannon Drop |
> | :--- | :--- | :--- | :--- |
> | 1 | 84.58 | 73.40 | 87.26 |
> | 3 | 85.77 | 81.56 | 87.92 |
> | 5 (Ours) | **86.94** | 81.55 | 88.26 |
> | 10 | 85.93 | **83.33** | **90.75** |
> | All | 55.43 | 65.66 | 74.81 |
>
> (b) Effect of EMA Span ($M=5$):
>
> | $EMA$ | LN-S Drop | LogTokU Drop | Shannon Drop |
> | :--- | :--- | :--- | :--- |
> | 1 | 84.58 | 73.40 | 87.26 |
> | 3 | 85.77 | 81.56 | 87.92 |
> | 5 (Ours) | **86.94** | 81.55 | 88.26 |
> | 10 | 85.93 | **83.33** | **90.75** |
> | All | 55.43 | 65.66 | 74.81 |
>
> (c) Effect of Top-$K$ ($span=5, M=5$):
>
> | Top-$K$ | LN-S Drop |
> | :--- | :--- |
> | $K=2$ | 86.05 |
> | $K=5$ | 85.42 |
> | $K=10$ | **87.64** |
> | $K=20$ (Ours)| 86.94 |
>
> Since $K$ has minimal impact, we use LN-S Drop as a representative example. Following Ma et al. [1], we default to $K=20$. Table (c) demonstrates high robustness across $K \in [2, 20]$, proving our method captures intrinsic dynamics without sensitive tuning
>
> The results demonstrate exceptional hyperparameter robustness. Across $M$, EMA span, and Top-$K$, performance remains highly stable within broad, non-extreme windows. Degradation occurs only in extreme settings that inherently violate our theoretical premises (e.g., $M \\rightarrow \\text{All}$ dilutes local signals via global averaging; span=19 erases topological drops via over-smoothing).
>
> Response to the questions:
>
> We clarify that our assumption $P_{\text{train}}=0$ for non-adjacent transitions models the idealized training corpora, not the LLM's generative behavior. Coherent human texts clear, continuous deductive chains ($A \rightarrow B \rightarrow C$) but rarely jump directly ($A \rightarrow C$). Thus, disjoint logical leaps are out-of-distribution (OOD) to some extent. This "non-adjacent transitions" is a well-established property that explains why LLMs need step-by-step reasoning (Prystawski et al., NeurIPS 2023) [2]. And I'm inspired by this and the relation is:
>
> Because the Evidence Manifold $E$ is an abstract topological space, its true distribution is intractable---similar to how the true $P(x)$ in Diffusion Models cannot be directly observed. Thus, we must rely on token-level empirical proxies.
> Operating at the token level is a deliberate necessity. Reasoning steps vary widely in length, making step-level metrics imprecise. By evaluating every token and applying Exponential Moving Average (EMA) smoothing, we form a continuous trajectory. The EMA can filter out some vocabulary noise (e.g., punctuation) while isolating the exact "weakest link" where the important reasoning collapse originates.
>
> That's why I think the assumption $P_{\text{train}}=0$ for non-adjacent transitions make sense.However,the assumption $P_{\text{train}} \sim P_{\text{true}}$ for adjacent transitions  is much more natural.
>
>
> Reference
>
> [1] Ma, H., et al. "Estimating LLM uncertainty with evidence." arXiv preprint arXiv:2502.00290 (2025).
>
> [2] Prystawski, B., Li, M., & Goodman, N. (2023). Why think step by step? reasoning emerges from the locality of experience. Advances in Neural Information Processing Systems (NeurIPS), 36, 70926-70947.

---

> > ### Author Rebuttal · Reviewer_CdZc · 2026-04-06
> >
> > My concerns have been adequately addressed and fully addressed. I've decided to maintain the score towards acceptance.

---

> > > ### Author Response · Authors · 2026-04-07
> > >
> > > We sincerely thank you for your time reviewing our rebuttal and for your positive feedback.

---

### Decision · Program_Chairs · 2026-04-30

**Decision:**

Accept (regular)

**Comment:**

The paper uses Evidence Drop to find hallucinations in reasoning. It tracks logic jumps using token entropy. Reviewers like that it needs no training. The authors added tests on logic data and big models during the discussion. Two reviewers have doubts about the test setup and the term hallucination. The other two reviewers like the strong results on math tasks. The consensus is that local drops beat global averages for finding errors. This work offers a simple way to check reasoning steps.